

# How do modeling choices impact the representation of
# structural connectivity and the dynamics of suspended
# sediment fluxes in distributed soil erosion models?
Magdalena Uber[1], Guillaume Nord[1], Cédric Legout[1], Luis Cea.[2]
[1]Univ. Grenoble Alpes, CNRS, IRD, Grenoble INP, IGE, 38000 Grenoble, France
[2]Environmental and Water Engineering Group, Department of Civil Engineering, Universidade da Coruña, A
Coruña,
*Correspondence to:* Cédric Legout (cedric.legout@univ-grenoble-alpes.fr) France
**1.Abstract**
Soil erosion and suspended sediment transport understanding is an important issue in terms of soil and water
resources management in the critical zone. In mesoscale watersheds (>10km²) the spatial distribution of potential
sediment sources within the catchment associated to the rainfall dynamics are considered as the main factors of
the observed suspended sediment flux variability within and between runoff events. Given the high spatial
heterogeneity that can exist for such scales of interest, distributed physically based models of soil erosion and
sediment transport are powerful tools to distinguish the specific effect of structural and functional connectivity on
suspended sediment flux dynamics. As the spatial discretization of a model and its parameterization can crucially
influence how structural connectivity of the catchment is represented in the model, this study analyzed the impact
of modeling choices in terms of contributing drainage area (*CDA*) threshold to define the river network and of
Manning's roughness parameter (*n*) on the sediment flux variability at the outlet of two geomorphological distinct
watersheds. While the modelled liquid and solid discharges were found to be sensitive to these choices, the patterns
of the modeled source contributions remained relatively similar when the CDA threshold was restricted to the
range of 15 to 50 ha, *n* on the hillslopes to the range 0.4-0.8 and to 0.025-0.075 in the river. The comparison of
both catchments showed that the actual location of sediment sources was more important than the choices made
during discretization and parameterization of the model. Among the various structural connectivity indicators used
to describe the geological sources, the mean distance to the stream was the most relevant proxy of the temporal
characteristics of the modelled sedigraphs.
**2.Introduction**
Soil erosion and suspended sediment transport are natural processes that can be exacerbated by human activities
and are thus a major concern for soils and water resources management. They cause on- and off-site effects such
as the loss of fertile top soil, muddy flooding, freshwater pollution due to the preferential transport of adsorbed
nutrients and contaminants, increased costs for drinking water treatment, reservoir siltation and aggression of fish
respiratory systems (Owens et al., 2005; Brils, 2008; Boardman et al., 2019). Although these problems are already
important in the Mediterranean and mountainous context (Vanmaercke et al., 2011), questions arise about the
future evolution of suspended sediment yields due to the expected increase on the intensity and frequency of severe



precipitation events in the following decades in these areas (Alpert et al., 2002; Tramblay et al., 2012; Blanchet et
al., 2018).
In mesoscale catchments (<100 km²), which correspond to a relevant scale for decision makers, correct modeling
of the hydrosedimentary responses requires a good understanding of the interactions between the spatiotemporal
dynamics of the rainfall with the spatial distribution of the catchment geomorphological characteristics. Several
studies have shown that the contributions of potential sediment sources can differ considerably from one flood
event to another and at different times of sampling within a flood event (Brosinsky et al., 2014 ; Gourdin et al.,
2014; Cooper et al., 2015; Gellis and Gorman Sanisaca, 2018; Vercruysse and Grabowski, 2019), particularly in
Mediterranean and mountainous watersheds (Evrard et al., 2011 ; Navratil et al., 2012; Poulenard et al., 2012;
Legout et al., 2013; Uber et al., 2019). Possible reasons for the observed variability of suspended sediment fluxes
from one event to another include seasonal variations of the climatic drivers of soil erosion and sediment transport,
variability of the spatial distribution of rainfall, land cover changes and human interventions (Vercruysse et al.,
2017).  At the event scale, the distribution of sources within the catchment and thus different travel times of
sediment from sources to the outlet as well as rainfall dynamics are assumed to be the dominant reason for the
observed suspended sediment flux variability (Legout et al., 2013).
Thus, the dynamics of suspended sediment fluxes during one event are hypothesized to result from the interplay
of structural and functional connectivity of the sources in the catchment. Wainwright et al. (2011) define structural
connectivity as the "extent to which landscape units are contiguous or physically linked to one another". What
makes up these landscape units depends on the scale and the study objectives. Structural connectivity can be
measured using indices of contiguity (Heckmann et al., 2018). It is an intrinsic property of the landscape, that
usually does not consider interactions, directionality and feedbacks. Functional connectivity on the other hand,
specifically describes the linkage of landscape units by processes that depend e.g. on the characteristics of rain
events. While some recent studies have shown the benefits of using the concepts of structural and functional
connectivity to understand the spatial and temporal variability of sediment fluxes (Cossart et al., 2018; Lopez-
Vicente and Ben-Salem, 2019), distinguishing both concepts remains challenging (Wainwright et al., 2011).
Distributed physically based models of soil erosion and sediment transport are powerful tools to distinguish the
specific effect of structural and functional connectivity on suspended sediment flux dynamics. Some recent studies
have already combined erosion and sediment transport modeling with sediment fingerprinting data (Theuring et
al., 2013; Wilkinson et al., 2013; Palazón et al., 2014, 2016; Mukundan et al., 2010a, 2010b). However, all of these
studies focused on long term mean source contributions, without working at high temporal resolution to understand
the dynamics of suspended sediment fluxes within and between flood events. Yet, numerical models can help to
understand the effect of the distribution of sources within the catchment, their linkage to the outlet, their travel
times and the characteristics of the rain events on the variability of suspended sediment source contributions
observed at the outlet.
The fact is that modeling soil erosion and sediment transport remains a challenge as there is no optimal model to
represent all erosion and hydrological processes in the catchment and there is no standard protocol for the choice
and set-up of the model (Merrit et al., 2003; Wainwright et al., 2008). Indeed, the outputs of hydro-sedimentary
models are very sensitive to choices made by the modeler in the way that processes are selected and spatially
implemented, as well as during model discretization, parametrization, forcing and initialization. We consider
especially that the spatial structure and the discretization of the model, as well as its parameterization can crucially



influence how structural connectivity of the catchment is represented in the model. In mesoscale catchments, the
connectivity of sources to the outlet depends a lot on the distance to the stream. In many cases, however, the
definition of the stream is not unambiguous (Tarboton et al., 1991, Turcotte et al., 2001). In most cases, the river
network is based on topographic analysis in GIS software, where a stream is made up of all the cells of the digital
elevation model (DEM) that exceed a threshold of contributing drainage area (CDA, Tarboton et al., 1991;
Colombo et al., 2007). The CDA of a DEM cell is the cumulative size of all cells that are located upstream of the
given cell and that drain into that cell. Thus, the definition of the stream and in consequence the connectivity of
active erosion sources to the outlet is highly dependent on the choice of the CDA threshold (Colombo et al., 2007).
Concerning parameterization, travel times of the sources to the outlet and thus structural connectivity also depend
on how surface water and sediment fluxes are calculated and parameterized. Many distributed models use the
depth-integrated shallow water equations (St. Venant equations) or different approximations of them, as the
kinematic or the diffusive wave approximations, for routing surface water to the outlet of the catchment (Pendey
et al., 2016). These equations are highly sensitive to the roughness parameter, which values depend whether
shallow water with partial inundation on hillslopes or concentrated flow in rivers are modelled (Baffaut et al.,
1997; Tiemeyer et al., 2007; Fraga et al., 2013, Cea et al., 2016). This paper contributes to improve our
understanding of the hydrosedimentary processes leading to sediment flux variability. We focus on the role of
structural connectivity using a distributed physical based model, applied to two mesoscale Mediterranean. Since
model outputs are supposed to be highly sensitive to the choices made during model discretization and
parameterization, the first objective is to assess the impact of these choices on the representation of structural
connectivity. A second objective is to assess how structural connectivity in turn impacts modeled suspended
sediment flux dynamics for both catchments.
**3. Methods**
**3.1. Characteristics of the modeled study sites**
Both study sites are long term research observatories belonging to the French network of critical zone observatories
(OZCAR, Gaillardet et al., 2018).
The 42 km$^2$ Claduègne catchment is a tributary of the Auzon river in Southeastern France. Being part of the
Cévennes-Vivarais Mediterranean Hydrometeorological Observatory (OHMCV, Boudevillain et al., 2011), the
catchment is a research site dedicated to the investigation of meteorological and hydrosedimentary processes
during heavy rain events and flash floods (Braud et al., 2014; Nord et al., 2017). The climate is dominated by
Mediterranean and oceanic influences with heavy rain events occurring mostly in autumn and to a lesser extent in
spring, and localized thunderstorms occurring more rarely in summer. These intense rain events can cause flash
floods and high sediment export. Average annual precipitation is 1050 mm (Huza et al., 2014). The geology of the
catchment is composed of basalts in the northern part and sedimentary rocks in the southern part. Uber et al. (2019)
identified three sources of suspended sediment: i) marly calcareous badlands are the major source of suspended
sediments due to their erodibility and connectivity to the river network, ii) diffuse sources on basaltic geology
comprising cultivated fields (mainly cereals) that are temporarily bare and iii) diffuse sources on sedimentary
geology equally comprise cultivated fields (mainly cereals) and vineyards where bare soil is found in between the
rows of the vine plants (Figure 1a). Table 1 gives the surface and the slopes of the catchment and the erosion zones.



The 20 km² Galabre catchment is a headwater catchment of the Bléone river located in the southern French alps
(Figure 1b). It is part of the Draix-Bléone Observatory dedicated to the study of hydrology and erosive processes
in a mountainous context with extensive badlands. The climate of the Galabre catchment, whose altitude varies
between 735 and 1909 m, is impacted by Mediterranean and mountainous influences with a mean annual
precipitation of around 1000 mm. There is a high seasonality with most precipitation occurring in spring and
autumn, although thunderstorms with high rain intensity also occur in summer (Esteves et al., 2019). The
catchment is entirely located on sedimentary rocks comprising limestones (34%), marls and marly limestones
(30%), gypsum (9%), molasses (9%) and Quaternary deposits (18%). A prominent feature of the catchment are
the badlands, that are found on all five types of rock and cover about 9.5% of the surface of the catchment (Esteves
et al., 2019). The land use is dominated by forests and scrublands, while agricultural zones are barely present in
the catchment. Suspended sediment fingerprinting studies revealed that most of the sediments originate from the
badlands of molasses and marls (Poulenard et al., 2012; Legout et al., 2013). Table 1 gives the characteristics of
the catchment.
Liquid and solid fluxes are continuously monitored at the outlets of both catchments with the same sensors and
protocols, from which suspended sediment yields are calculated (Table 1). Water level is measured with an H-
radar and converted to discharge with a stage discharge rating curve. Suspended sediment concentrations are
monitored with turbidimeters and suspended sediment samples are automatically taken every 40 min once a
threshold of turbidity and water level is exceeded. These samples are dried and weighed and are used to establish
a rating between turbidity and suspended sediment concentrations.
In order to quantify the structural connectivity of the sources in the catchments, four indicators were calculated,
i.e. the distance to the outlet, distance to the stream and the two indices of connectivity (IC) proposed by Borselli
et al. (2008) and Cavalli et al. (2013). Maps of the distance to the outlet along the flowlines (i.e. the distance that
water and sediments travel following the gradient of the terrain elevation) and the distance to the stream were
created. For the latter, the stream network obtained with a CDA threshold of 50 ha was used. The distance to the
outlet and the distance to the stream of a given position in the catchment serve as proxies of longitudinal and lateral
connectivity in the sense of Fryirs (2013). Both maps were created using TauDEM (Tarboton, 2010) and a digital
elevation model at a resolution of 1m (Claduègne: bare earth Lidar DEM, Nord et al., 2017; Galabre: RGE ALTI
product of IGN, 2018). However, neither of these measures takes into account surface roughness and slope. Thus,
two of the most widely used indicators of connectivity, i.e. the IC proposed by Borselli et al. (2008) and the
adjusted version of IC proposed by Cavalli et al. (2013), were calculated. Both indicators were calculated for each
pixel of the DEM and take into account the CDA of that pixel and the distance to the stream along the flow lines.
They also both include a weighting factor for the mean slope in the CDA and along the downstream path as well
as a second weighting factor $W$. Borselli et al. (2008) weight the index with land use, thus the factor $W$ was derived
from the values proposed by Panagos et al. (2015) for the land use data that was obtained from Inglada et al.
(2017). Cavalli et al (2013) on the other hand propose a roughness index as the weighting factor $W$ that represents
a local measure of topographic surface roughness that is calculated for a 5 x 5 cell moving window. Both indicators
were calculated using the program SedInConnect (Crema and Cavalli, 2017). All these four indicators were
calculated for each pixel within the catchments and their values on the erosion zones were extracted. Mean values
and standard deviations are given in Table 1, while the distributions of the distance to the outlet and to the stream

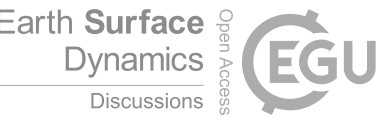

are shown in Figure 2. These characteristics of the catchments indicate that not only erodibility but also structural
connectivity differs strongly between the two catchments and between sources.

**3.2.Model description**
Equations describing the hydraulic routing of water, soil erosion and sediment transport are implemented in the
2D software Iber (Cea and Bladé, 2015):
*Hydrodynamic module*
Water depth and velocity fields are derived from the solution of the full St. Venant equations applied both on the
hillslopes and in the river network. Including rainfall and infiltration terms as well as Manning's formula for bed
friction they can be written as:
$$\frac{\partial h}{\partial t} + \frac{\partial q_x}{\partial x} + \frac{\partial q_y}{\partial y} = R - I$$

$$\frac{\partial q_x}{\partial t} + \frac{\partial}{\partial x}\left(\frac{q_x^2}{h}\right) + \frac{\partial}{\partial y}\left(\frac{q_x q_y}{h}\right) = -gh\frac{\partial z_s}{\partial x} - gh\frac{n^2}{h^{7/3}}|q|q_x \qquad (1)$$

$$\frac{\partial q_y}{\partial t} + \frac{\partial}{\partial x}\left(\frac{q_x q_y}{h}\right) + \frac{\partial}{\partial y}\left(\frac{q_y^2}{h}\right) += -gh\frac{\partial z_s}{\partial y} - gh\frac{n^2}{h^{7/3}}|q|q_y$$

where $h$ is water depth, $t$ is time, $q_x$ and $q_y$ are the components of unit discharge in the two horizontal directions, $R$
is rainfall intensity, $I$ is the infiltration rate, $g$ is gravity acceleration, $z_s$ is the elevation of the free surface and $n$ is
Manning's roughness parameter. As the focus of this study is on choices made during model set-up and how
structural connectivity is represented, a synthetic triangular hyetograph (duration of 12 h, maximum intensity of 5
mm h[-1]) representing effective precipitation (i.e. $R$-$I$) is applied spatially homogeneous over the entire catchment.
*Soil erosion module*
The full description of the soil erosion model can be found in Cea et al. (2016) and a summary is given here. The
complete soil erosion model uses a two-layer soil structure that consists of one layer of eroded material over a
layer of non-eroded cohesive soil. Given the results of Cea et al. (2016) that the two-layer structure of the model
increases its complexity without significantly improving its predictive capacity in real applications, we only use a
single-layer structure with vertically uniform erodibility. We assume that the single-layer structure is adequate for
the badlands where there usually is a thick regolith layer, and erosion from the underneath cohesive layer is
negligible compared to the one of the regolith layer. In the complete model, two particle detachment processes are
considered, i.e. rainfall-driven detachment and flow-driven entrainment. In our case, we assume that rainfall-driven
detachment is the most significant of both processes and thus, it is the only detachment mechanism considered in
our simulations. We further assume that all eroded particles are transported in suspension to the outlet and that
deposition is negligible. This wash load hypothesis leads to a further simplification of the erosion module compared
to the original one proposed by Cea et al. (2016), i.e. the omission of the deposition term. Thus, the suspended
sediment concentration at every time step and location is calculated from Eq. 2, which is a simplified version of
the equation given in Cea et al. (2016) for the case where a single-layer structure, only rainfall-driven detachment
and no deposition are assumed:


$$\frac{\partial hC}{\partial t} + \frac{\partial q_x C}{\partial x} + \frac{\partial q_y C}{\partial y} = D_{rdd} \qquad (2)$$

where C [kg m$^{-3}$] is the depth-averaged sediment concentration in the water column and D$_{rdd}$ [kg m$^{-2}$ s$^{-1}$] is the
rainfall-driven detachment rate that is calculated assuming a linear relationship between the detachment rate and



the rain intensity, i.e. $D_{rdd} = \alpha R$, where $\alpha$ [kg m$^{-3}$] is the rainfall erodibility coefficient that represents the flux of
sediment mass detached per unit area by a unit rainfall intensity.
*Solution schemes*. The model equations are solved with a finite volume solver, using an explicit temporal
discretisation. A detailed description of the numerical schemes is beyond the scope of this paper. The reader is
referred to Cea and Bladé (2015) and Cea and Vázquez-Cendón (2012) for details on the numerical methods.

**3.3. Model discretization and input data**
The geometry of the catchments is divided in three main modeling units with different spatial discretizations and
roughness coefficients, i.e. the river network, the hillslopes and the badlands. The river bed was delineated by i)
identifying the river network using TauDEM (Tarboton, 2010) and ii) creating a polygon by "buffering" the line
feature of the river. In order to take into account that the width of the river varies from upstream to downstream,
we introduced a distinction between the perennial river network defined using a CDA of 500 ha and the intermittent
river network obtained using a CDA of 15 ha. While the highest value of 500 ha is often used for cartography and
large scale modeling studies (e. g. Colombo et al., 2007; Vogt et al., 2007; Bhowmik et al., 2015), the smallest
value of 15 ha was found to create a river network that includes the intermittent streams observed in the catchment.
For the former a buffer of 10 m to both sides of the river was applied. For the latter, composed of small tributaries
and in good agreement with field observations of the whole extension of the hydrographic network during floods,
a buffer of 5 m was applied. The badlands were delineated based on orthophotos and verified during field trips,
while the hillslopes cover the rest of the catchments.
These principal modeling units were discretized as a finite volume mesh. In our study, we used an unstructured
triangular mesh with variable mesh size in the different units. The smallest mesh size was required in the river
network, where water and sediment fluxes are concentrated, so it was set to 5 m. On the hillslopes a coarser mesh
size of 100 m was chosen in order to reduce the number of elements and thus computation time. In the badlands,
where the fluxes are concentrated in the steep gullies, an intermediate mesh size of 20 m was used. At the border
between two landscape units the mesh size evolves gradually. With this discretization the model of the Claduègne
consists of roughly 173.000 mesh elements, while the one of the Galabre catchment of 75.000 elements. The
roughness coefficients were spatially uniform in each modeling unit but could vary from one scenario to another
with values ranging from 0.025 to 0.1 in the river and from 0.2 to 0.8 in the two other units.
While equations 1 and 2 are solved on the entire catchment, the production of sediments was restricted to the
potential erosion sources that were classified according to i) their geology, i.e. in three classes for the Claduègne
and four for the Galabre catchment (Figure 1), ii) their geology and their distance to the outlet (Figure 2a,c) and
iii) their geology and their distance to the stream network (Figure 2b,d). Separate sedigraphs were calculated for
each source class, solving equation 2 in each mesh element for each source class separately. The rain erodibility
coefficient $\alpha$ of each geological class was estimated from the available observed time series of suspended sediment
concentrations (SSC), discharge and rainfall. Using the discharge and SSC, the suspended sediment flux was
calculated and integrated over time for each recorded event to obtain event suspended sediment yield SSY$_{ev}$ [g].
The value of $\alpha$ was estimated separately for every event and every source as:
$$\alpha_{s,ev} = \frac{SSY_{s,ev}}{R_{ev} \cdot A_s} \qquad (3)$$
where $A_s$ is the erodible surface of the respective source and $R_{ev}$ [mm] is the amount of effective rainfall during
the respective event. SSY$_{s,ev}$ is the contribution of source $s$ to SSY$_{ev}$ and was calculated based on the mean source

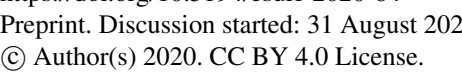



contributions obtained from sediment fingerprinting studies in the Claduègne (Uber et al. 2019) and the Galabre
(Legout et al., 2013). An average value of $\alpha_s$ [g mm$^{-1}$ m$^{-2}$] was calculated by averaging over all the available
observed events (Table 1).

**3.4. Modeling scenarios**
In order to test the effects of model discretization and parameterization on the representation of structural
connectivity and on the computed suspended sediment fluxes, the modelling scenarios shown in Table 2 were
tested.
*Sc.1: Basic scenario*
In the basic scenario the threshold to define the river network was set to 15 ha and the sources were classified
according to their geology as in the sediment fingerprinting studies. In the river network units, Manning's *n* was
set to 0.05 and in the hillslopes and badlands units it was set to 0.8. The value in the river network corresponds to
what can be expected from values reported in the literature for streams comparable to the Claduègne and the
Galabre (Te Chow, 1959; Barnes, 1967; Limerinos, 1970). For the values on the hillslopes there are fewer
recommendations from the literature as the use of the St. Venant equations for the calculation of fluxes on
hillslopes is much less common. Existing studies indicate that the values have to be considerably higher than those
used commonly in river flow models (Engman et al., 1986; Hessel et al., 2003; Fraga et al., 2013; Hallema et al.,
2013). As these values are uncertain, the impact of this parameterization was assessed in further scenarios. The
basic scenario was used as the main reference to compare the other scenarios to and for the comparison between
the two catchments.
*Sc. 2: Impact of the CDA threshold*
We tested the impact of varying the CDA threshold on the modeled hydro-sedimentary response while keeping all
other parameters unchanged compared to the basic scenario. As different values for Manning's n were applied in
the river network units on one hand and in the hillslopes and badlands units on the other hand, the travel times of
the sediments from source to sink vary depending on the length of the river network in the model. Five values of
the CDA threshold were used: 15, 35, 50, 150 and 500 ha.
*Sc. 3: Impact of the parameterization of Manning's n*
As one of the objectives of this study is to assess the impact of choices made during model set-up on the simulated
sediment flux dynamics, the model was run with different values of Manning's *n* in the river network units on one
hand and in the hillslopes and badlands units on the other hand. In the river network units, values were varied
spanning a range from 0.025 to 0.100. This corresponds to the full range of plausible values (Te Chow, 1959;
Barnes, 1969; Limerinos, 1970). In the hillslopes and badlands units, the value of 0.8 used in the basic scenario is
already at the upper end of values reported in the literature (e.g. Te Chow, 1959; Engman, 1986; Hessel et al.,
2003; Hallema et al., 2013). Thus, values in the range of 0.2 to 0.8 were tested.
*Sc. 4: Source classification based on connectivity*
In order to test how the spatial distribution of the sources in the two distinct catchments contribute to the modeled
sedigraph at the outlet, the geological sources were classified into subclasses based on their distance to the outlet
(Sc 4a,c) and distance to the stream (Sc 4b,d). These two measures serve as a proxy for the structural connectivity
of the sources. The underlying hypothesis is that depending on their connectivity, several patches of the same
source have different travel times to the outlet and can therefore lead to several peaks in the sedigraph of the



source. In Sc 4b and 4d , the geological sources were classified in two groups based on their distance to the stream.
The badland sources in both catchments were classified as being directly adjacent to the stream network or not.
The diffuse sources in the Claduègne catchment, i.e. soils on basaltic and sedimentary geology, were classified
using a threshold of distance to the stream of 150 m. In Sc 4a and 4c, the geological sources were classified in one
to four groups depending on their distribution to the outlet (Figures 2a and 2c).

**3.5.Comparison of scenarios**
Modelled outputs for each scenario can be accessed and visualized through Uber et al. (2020). To assess the impact
of the changes done in each scenario with respect to the basic scenario, several characteristics of the modeled
hydrograph and sedigraphs of all sources were calculated. The lag time of liquid discharge $T_{lag,Ql}$ is calculated as
the time between the barycenter of the hyetograph and the barycenter of the hydrograph. The time of concentration
of liquid discharge $T_{c,Ql}$ is defined as the time between the end of effective precipitation and the end of the outlet
hydrograph. A third characteristic time, $T_{spr;Ql}$, was defined to assess the spread of the hydrograph and thus, a
characteristic duration of the flood event (Figure 3). All of these measures were also calculated for solid discharge
($T_{lag,Qs}$, $T_{c,Qs}$, $T_{spr,Qs}$) and for each source separately. Further, maximum liquid discharge $Q_{l,max}$ and solid discharge
$Q_{s,max}$ were determined for each scenario. Our simulations were truncated 12 h after the end of precipitation and in
some cases fluxes did not recede to zero, so a threshold of 0.1 $Q_{max}$ was used to calculate $T_{lag}$, $T_c$ and $T_{spr}$ for solid
and liquid discharges.

**4.Results and discussion**
**4.1.Impact of modeling choices on modeled sediment dynamics**
*Varying the contributing drainage area threshold*
Results show that the model was sensitive to the choice of the CDA threshold used to define the river network.
Figure 4 shows the modeled hydrographs that were obtained when the CDA threshold was varied from 15 to 500
ha. For both catchments, higher values led to a less steep rising limb of the hydrograph, lower and later peak flow,
slower recession and a flatter hydrograph (Figure 4a,c). Thus, the lag time $T_{Lag}$, time of concentration $T_c$ and time
of spread $T_{spr}$ of liquid discharge increased with increasing CDA threshold (Figure 5a,b,c; Table 3). In both
catchments, the hydrographs obtained with thresholds of 15, 35 and 50 ha were relatively similar, but the results
obtained with 150 and 500 ha differed considerably. In the Claduègne catchment peak flow was reduced by
approximately a factor 2 when the threshold was increased from 15 to 500 ha, while in the Galabre catchment it
decreased by about 20% (Table 3). In the Claduègne catchment the hydrograph obtained with the threshold of 500
ha was much flatter than the one in the Galabre catchment and the recession was very slow, so that even 12 h after
the end of precipitation, discharge at the outlet persisted. This was not the case in the Galabre catchment.
The different hydrological response could not be attributed to the difference in size of the catchments alone,
because a subcatchment of the Claduègne that has the same size as the Galabre catchment and a similar mean slope
than the entire Claduègne catchment (mean +/- sd: 25 +/- 32 %) also had a less steep rising limb of the hydrograph
than the Galabre (Figure 4b). The $T_{Lag}$ of 3.2 h (basic scenario) was smaller than the one of the Claduègne
catchment at the outlet (4 h) but also considerably larger than the one of the Galabre catchment (2.3 h). Thus, we
assume that the fast rise and recession of the hydrograph in the Galabre catchment were mainly due to the steeper
slopes in this catchment (Table 1) given that the lengths of the river networks are similar.





The modeled response of the sedigraphs were also very sensitive to the CDA threshold. $T_{lag}$, $T_c$ and $T_{spr}$ of solid
discharge increased generally with increasing CDA threshold, in particular from 150 to 500 ha (Figure 5a,b,c;
Table 3). Nevertheless, the changes of CDA did not affect the sedigraphs similarly for each sediment source. In
the Claduègne catchment, the sedigraphs obtained with CDA thresholds of 15, 35 and 50 ha were similar to each
other, but when larger values were used, they varied substantially for each sediment source (Figure 6a,b,c,d). In
particular, the sedigraphs of the basaltic and sedimentary sources were considerably delayed when the 500 ha
threshold was used. In the Galabre catchment the sedigraphs of all sources were highly sensitive to significant
changes of the CDA threshold with changes in $T_{lag,Qs}$ and $T_{c,Qs}$ of more than 100% for the CDA threshold of 500ha
(Table 3). When the threshold of 500 ha was used, the shape of the sedigraph of some sources differed. Indeed,
for the badlands in the Claduègne catchment and the black marls and the molasses in the Galabre catchment, the
single peak sedigraph turned into a multi peak sedigraph (Figure 6).
The differences in the modeled sedigraphs when different values for the CDA threshold were used were also
obvious when the simulated contributions of the sources to total suspended sediment load were regarded (Figure
7 and interactive figures at https://shiny.osug.fr/app/EROSION_MODEL.2020). Increasing the CDA threshold
from 15 to 500 ha notably prolonged the first flush of black marl dominated sediment in the Galabre catchment
(marked as "1" in Figure 7c,d). During the rising limb of the hydrograph and peak flow (marked "2"), the source
contributions were variable while they remained relatively constant during the recession period ("3") when the
CDA threshold of 500 ha was used. This was not the case when the threshold was set to 15 ha. In this case, the
contribution of molasses decreased steadily throughout the event while the one of limestone and quaternary
deposits increased ("2","3", and "4" in Figure 7c). In the Claduègne catchment notably the arrival of the basaltic
sources at the outlet was much delayed when the CDA threshold of 500 ha was used compared to when the one of
15 ha was used. The shape of the sedigraph with multiple peaks that was modeled with a threshold of 500 ha
resulted in a slower and less steady recession of the badland sources (Figure 7b).
Overall, our results showed that the thresholds of 15, 35 and 50 ha produced very similar results, i.e. the catchments
were not very sensitive to the CDA threshold in this range. The parameters given in Table 3 changed by a maximum
of 37% compared to the basic scenario. Other authors have shown that the CDA thresholds can vary spatially (i.e
different values are found in different subcatchments) and temporally (CDA thresholds vary between seasons or
between events; Montgomery et al., 1993; Bischetti et al., 1998; Colombo et al., 2007). In the studied catchments,
variability in this range seemed not to be of prime importance. However, the larger thresholds of 150 and 500 ha
changed the modeled sediment dynamics considerably (changes of up to 280% with respect to the basic scenario
and several parameters changed > 150%, Table 3). This result showed that it is important to use a CDA threshold
that is in the right order of magnitude compared to field observations or detailed maps (i.e. topographic map at
scale 1:25000). Pradhanang and Briggs (2014) also tested the effect of CDA threshold on annual sediment yield
and streamflow modeled with the AnnAGNPS model. In their study, they observed a high sensitivity of the model
output to variations of the CDA threshold from 0.5 to 20% of catchment area (5-25 km²). Differently to our study,
they did not observe a convergence of the results in the "right" order of magnitude of the CDA threshold but results
differed strongly between the 6 considered catchments.




***Varying Manning's n***

Changing Manning's $n$ influenced the timing, the peak and the spread of both liquid and total solid discharge (Figure 8, Table 3). In general, increasing $n_{river}$ *and* $n_{hillsl.}$ led to a later time of rise of the hydrograph, a later time of peak and to slower recession with longer $T_{lag,Ql}$ and $T_{c,Ql}$ (Figure 5, Table 3). Nevertheless $Q_{lmax}$, $T_{lag,Ql}$, $T_{c,Ql}$ and $T_{spr,Ql}$ were less sensitive to changes of $n_{river}$ *and* $n_{hillsl.}$ in the Galabre than in the Claduègne catchment (Figure 5, Table 3). While increasing $n$ also led to less maximum liquid discharge, this was not the case for solid discharge. Peak solid discharge even increased with increasing $n_{river}$ in the Claduègne catchment and to a lesser degree also in the Galabre catchment (Table 3). Interestingly, in the Claduègne catchment liquid discharge was more sensitive to changes in $n_{hillsl.}$ than to $n_{river}$ while solid discharge was more sensitive to $n_{river}$. This was not the case in the Galabre where both liquid and solid discharges were more sensitive to $n_{hillsl.}$.

Changing Manning's n also influenced the temporal dynamics of source contributions. A low $n_{hillsl.}$ of 0.2 led to a multi-peaked sedigraph in the Claduègne catchment (Figure 8b). This difference in the shape of the sedigraph also led to a difference in the modeled temporal dynamics of the percentage of source contributions (Figure 9a). When $n_{hillsl.}$ was set to 0.2, the decrease of the contribution of the badland sources to total suspended sediment load in the Claduègne catchment was slower during the main part of the event (marked "2" in Fig 9a) and the break point between phase 2 and 3 in the decrease of the badland source was more pronounced than in the basic scenario where $n_{hillsl.}$ was set to 0.8 (Figure 7a). In fact, for several hours during phase 2, the contributions of the three sources were nearly constant. This was not the case for the scenarios 3b and 3c where $n_{hillsl.}$ was set to 0.4 and 0.6. These scenarios hardly differed from the basic scenario (see interactive figures). In the Galabre catchment the scenarios 3b and 3c also hardly differed from the basic scenario. When $n_{hillsl.}$ was set to 0.2, the contributions during the main part of the event ("2" in Figure 9b) remained more stable than in the basic scenario (Figure 7c).

Changing $n_{river}$ hardly changes the dynamics of the modeled source contributions in both catchments (see interactive figures). In the Claduègne catchment, increasing $n_{river}$ from 0.025 to 0.1 generally increased $T_{lag,Qs}$ and $T_{c,Qs}$ (Figure 5, Table 3) and led to a slight prolongation of the first flush of sediments from the sedimentary source. In the Galabre this was also the case for the first flush of sediments originating from black marl, as it was the case for the changes in the CDA threshold shown in figure 7d.

Our results showed that even though modeled liquid discharges were sensitive to $n_{hillsl}$, the sedigraphs of the main sources and thus of total suspended solid discharge were much less sensitive to this parameter (Figure 8). This was due to the fact that in both catchments the main sediment sources were located close to the river (Table 1, Figure 2). Thus, only a small fraction of the trajectory of particles was located on the hillslopes. This was also represented in the modeled dynamics of the source contribution which barely changed unless the most extreme value of 0.2 was applied. This result suggests that it is sufficient to have a rough idea of the value of Manning's $n$ to study the dynamics of sediment fluxes. In the Claduègne catchment the modeled sedigraph was affected by variations of $n_{river}$ which was less true for the Galabre catchment. This might be related to the difference of slopes of the river network in both catchments. Indeed, the mean slope in the river network is 2-3 times higher in the Galabre than in the Claduègne catchment (Table 1), suggesting that the model was more sensitive to changes in Manning's $n$ when slopes were low. However, also in the Claduègne catchment, changes in $n_{river}$ did not change the modeled dynamics of the source contributions, which was again encouraging for the use of this type of model to understand hydro-sedimentary dynamics.



### 4.2. The role of structural connectivity on the dynamics of suspended sediment fluxes at the outlet

The application of the same rainfall event with a similar spatial discretization and parameterization to the two studied catchments (i.e. basic scenario) allowed to provide a more detailed analysis on how their respective characteristics influenced their hydrosedimentary response. A first result was that the Galabre catchment reacted faster than the Claduègne catchment. The hydrographs and the sedigraphs rose earlier than in the Claduègne catchment. The rising limb of the hydrograph was also steeper in the Galabre than in the Claduègne catchment (shorter $T_{lag}$ and $T_c$, Figure 5, Table 3). We assume that this was mainly due to the steeper slopes of the Galabre catchment (Table 1). From Figures 7 and 9 a general pattern of the contribution of the different geological sources to total solid discharge can be derived. In the Claduègne catchment at the onset of the event ("1"), the sediments originated from the sedimentary source and the badlands. During the phases 2 and 3 of the event, the main source (i.e. the badlands, Table 1) clearly dominated total solid discharge. The contribution of this source decreased gradually while the percentage of contribution of the two others increased. In the Galabre catchment at the onset of the event ("1"), suspended sediment originated almost entirely from the black marls. In the second phase of the event, the main source (i.e. molasse) arrived and clearly dominated total solid discharge. Thereafter, the contribution of the molasses decreased while the one of the limestones and the quaternary deposits increased (phases 3 and 4). These general patterns were broadly consistent with the location of the different geological sources in the two catchments. However, some discrepancies appear when comparing the timings of arrivals of the various geological sources to the ranking of the various connectivity indicators (i.e. distance to stream, to outlet, $IC$ Borselli and $IC$ Cavalli). The lag times of the sources in the Claduègne catchment could generally be ranked as $T_{lag,Qs}\ bad < T_{lag,Qs}\ sed < T_{lag,Qs}\ bas$ (Table 3, Figure 5). This was also true for $T_{c,Qs}$ and $T_{spr,Qs}$ and consistent with the ranking of the mean distance to the stream as well as with both mean IC values but not with the mean distance to the outlet, as the sedimentary sources were the closest from the outlet (Table 1). In the Galabre catchment $T_{lag,Qs}$, $T_{c,Qs}$ and $T_{spr,Qs}$ of the molasses and marls were always smaller than the ones of quaternary deposits and limestones (basic scenario, Table 3). This was coherent with the ranking of mean distances to the stream but not with the ranking of mean distances to the outlet nor with the one of mean IC values (Table 1). Actually, the mean IC values in the Galabre were very similar for each of the four geological sources of sediments and could not really be used to discriminate the sources in terms of the timing of arrivals of the sedigraphs at the outlet.

To further address the respective roles of the distance to the outlet and the distance to the stream on the pattern of source contributions to total solid discharge throughout events, the geological sources were subdivided based on these measures in the scenarios 4a to 4b (Table 2). Figures 10 and 11 showed for the Galabre catchment that the limestone sources that were close to the river and the ones that were close to the outlet exhibited a clockwise hysteresis pattern while the distant ones exhibited an anticlockwise pattern. These results confirmed typical interpretations of discharge-sediment flux hysteresis (Bača, 2008; Misset et al., 2019) and highlighted that the sedigraphs of the different sediment sources were strongly related to their location in the catchments and their structural connectivity. The absence of coherent trends of the ranking of the $T_{lag,Qs}$ with the one of the mean distances of the sources to the outlet could be related to the distribution of the distances to the outlet of all sediment sources that were generally more scattered than the distribution of the distances to the stream, particularly for the Galabre catchments (Figures 2c,d). Thus, the mean distance to the outlet could not be fully representative of a



given geological source. Additionally, the triangular rain applied to both catchments lasted a rather long period,
much longer than the times of concentration of both catchments. Thus, the sedigraphs of all subsources were
stretched over a time span that was comparable to the time span of the rain event. The distant sources arrived at
the outlet long before the flux of the close sources ceased. Consequently, the sedigraphs of the different subsources
of both catchments were superposed and did not lead to separate peaks.
Even though different patches of closer and more distant subsources did not lead to multipeak sedigraphs and thus
to a very high flux variability, the classification into close and distant subsources from the outlet allowed to explain
the dynamics of source contributions. The first peak of black marls that arrived at the outlet of the Galabre during
the onset of the event, originated entirely from the subsources that were close to the outlet and adjacent to the river
network (marked "1" in Figures 10e and 11e). For the molasses and quaternary deposits, the distance to the river
or the outlet hardly impacted the variability of the predicted source contributions. The first molassic sediments that
arrived at the outlet during the rise of the hydrograph ("2"), originated almost entirely from the molassic patch that
was directly adjacent to the river network. However, the decrease of the contribution of the adjacent sources during
peak flow ("3") occurred simultaneously with the arrival of the further sources.
A similar dynamic was observed in the Claduègne catchment. The first flush of sediments with a high contribution
from the sedimentary source, originated entirely from sedimentary sources that were directly adjacent to the stream
and from the badlands that were closest to the outlet (marked "1" in Figures 12e and 13e). When the distance to
the outlet was considered, it was remarkable that sediments which originated from the class badland 3
(corresponding to a distance to the outlet of 7.5-10 km; $T_{lag,Ql}$ = 2.17 h) arrived during the rising limb of the
hydrograph ("2") before the ones that originated from badland 2 (distance to the outlet of 5-7.5 km, $T_{lag,Ql}$ = 2.67
h) even though they were further away from the outlet. This was coherent with the distance to the stream. While
all patches belonging to the class badland 3 were directly adjacent to the river network, the ones belonging to the
class badland 2 were further away from the river. It should however be stressed that this finding was related to the
parameterization of the model and the choice of using contrasted roughness coefficients in hillslopes and in the
river. In the results of scenario 4c where $n_{river}$ was set to 0.1 and $n_{hillsl.}$ was set to 0.2 (i.e. less difference between
$n_{river}$ and $n_{hillsl.}$) this was not observed.
The fact that in both catchments different hysteresis loops were observed for subsources of different connectivity
showed that the subsources exhibited different hydrosedimentary behavior. It also showed that even a simple
classification based on the distributions of the geological sources of sediments according to their distance to the
stream or the outlet could help to understand the sediment flux dynamics at the outlet of mesoscale catchments.
Among the various connectivity indicators (i.e. distance to stream, to the outlet, IC Borselli, IC Cavalli) tested in
both studied catchments, the mean distances of the various geological sources to the stream were the most robust
proxies of the rankings of the three temporal characteristics of sedigraphs (i.e. $T_{lag}$, $T_c$ and $T_{spr}$). Overall, our results
showed that the location of the sources in the catchment highly influenced the temporal dynamics of suspended
solid discharges at the outlet. The main characteristics of the sediment flux dynamics were observed for all the
modeling scenarios. While the two studied mesoscale catchments and also the subsources of sediments within the
same catchment exhibited different sensitivities to model discretization and parametrization, one main result of
this study was that the actual location of sediment sources and their structural connectivity were more important
than the modeling choices. Indeed, as soon as appropriate CDA thresholds (typically 15 to 30ha) and Manning's
$n$ (in streams typically between 0.03 and 0.06 and on hillslopes between 0.4 and 0.8) were used, the temporal



dynamics of the modeled contributions of the different sources were relatively independent of the modeling
choices. Values could be varied in quite a high range without significantly changing these flux dynamics. As this
finding could be different for different types of rain events, notably shorter events, further studies should focus on
the influence of rainfall dynamics on modelled sediment fluxes in mesoscale catchments.

**5.Conclusion**
This study aimed to improve our understanding of hydrosedimentary processes leading to variability in the
contribution of potential source soils to suspended sediments at the outlet of mesoscale catchments using a
distributed, physically based numerical model. It allowed to assess to which extent the modeling choices made
during model discretization and parameterization could impact the representation of the structural connectivity in
two mesoscale catchments. As structural connectivity represents the way sediment sources are topologically
connected to the catchment outlet we considered that the main elements to be considered were the location of the
sources with respect to the river network, the length between the point of entry of the source into the river network
and the outlet of the catchment, and the friction parameters that will interact with the slope to explain the temporal
distribution of sediment flows at the outlet.
We observed that the model was sensitive to the contributing drainage area threshold to define the river network
and to Manning's roughness parameter $n$ in the river network and on hillslopes. However, the model was less
sensitive to all three values once the parameters varied only in a restricted, reasonable range. In our study sites,
the pattern of modeled source contributions remained relatively similar when the CDA threshold was restricted to
the range of 15 to 50 ha, $n$ on the hillslopes to the range 0.4-0.8 and to 0.025-0.075 in the river. In both studied
catchments the actual location of sediment sources and their structural connectivity was found to be more important
than the choices made during discretization and parameterization of the model.
Comparing the two studied catchments showed that their hydrosedimentary responses differed due to the different
locations of the sources in the catchments and the slopes of the river network and hillslopes. Among the various
structural connectivity indicators used to describe the geological sources, the mean distance to the stream was
found to be the most relevant proxy of the temporal characteristics of the sedigraphs. Nevertheless, the
classification of the geological sources in subgroups according to the distance to the outlet and to the stream
allowed a better assessment of the timings of suspended sediments at the outlets.

**6.Acknowledgements**
The authors would like to acknowledge the Ciment platform of the Université Grenoble Alps for access to
calculation clusters, the Draix Bléone and OHMCV long term observatories funded by the National Institute of
Science of the Universe for access to data sets and the OZCAR research infrastructure. The authors are grateful to
Laurent Bourgès, Rémi Cailletaud and OSUG for the publication of the DOI of dataset and the deployment of
shinyproxy on the OSUG servers to host the interactive application that enables to visualize the dataset.

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



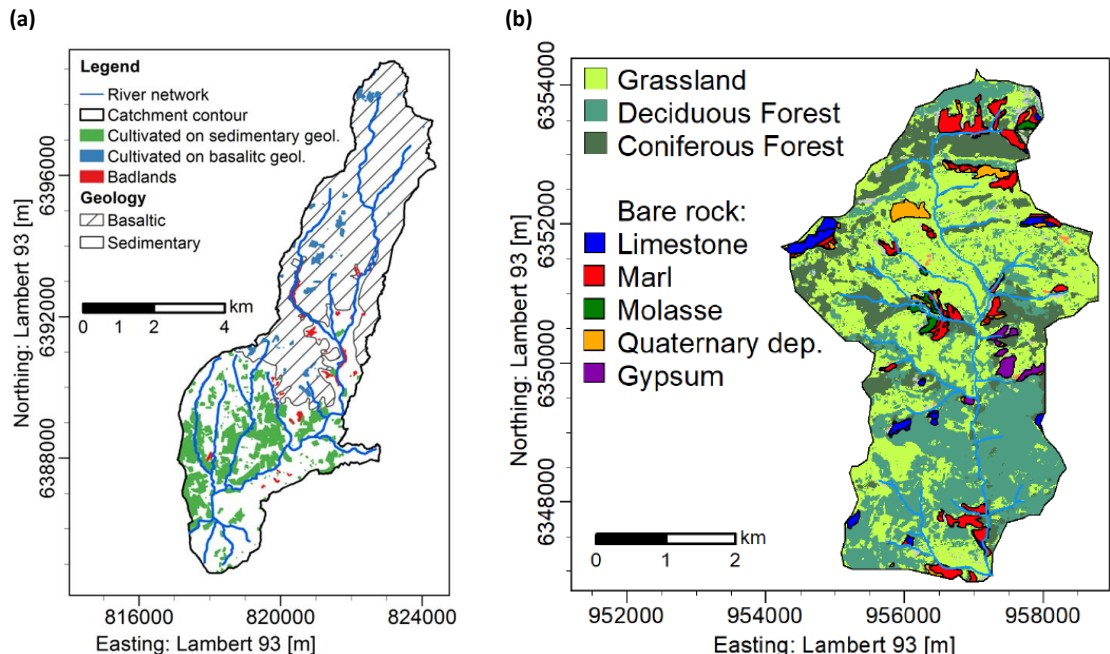


**Figure 1:** Maps of the (a) Claduègne and (b) Galabre catchments. Note that gypsum badlands are not considered
in this study as this material is highly soluble and do not contribute to sediment fluxes.





Earth **Surface**
Dynamics
Discussions

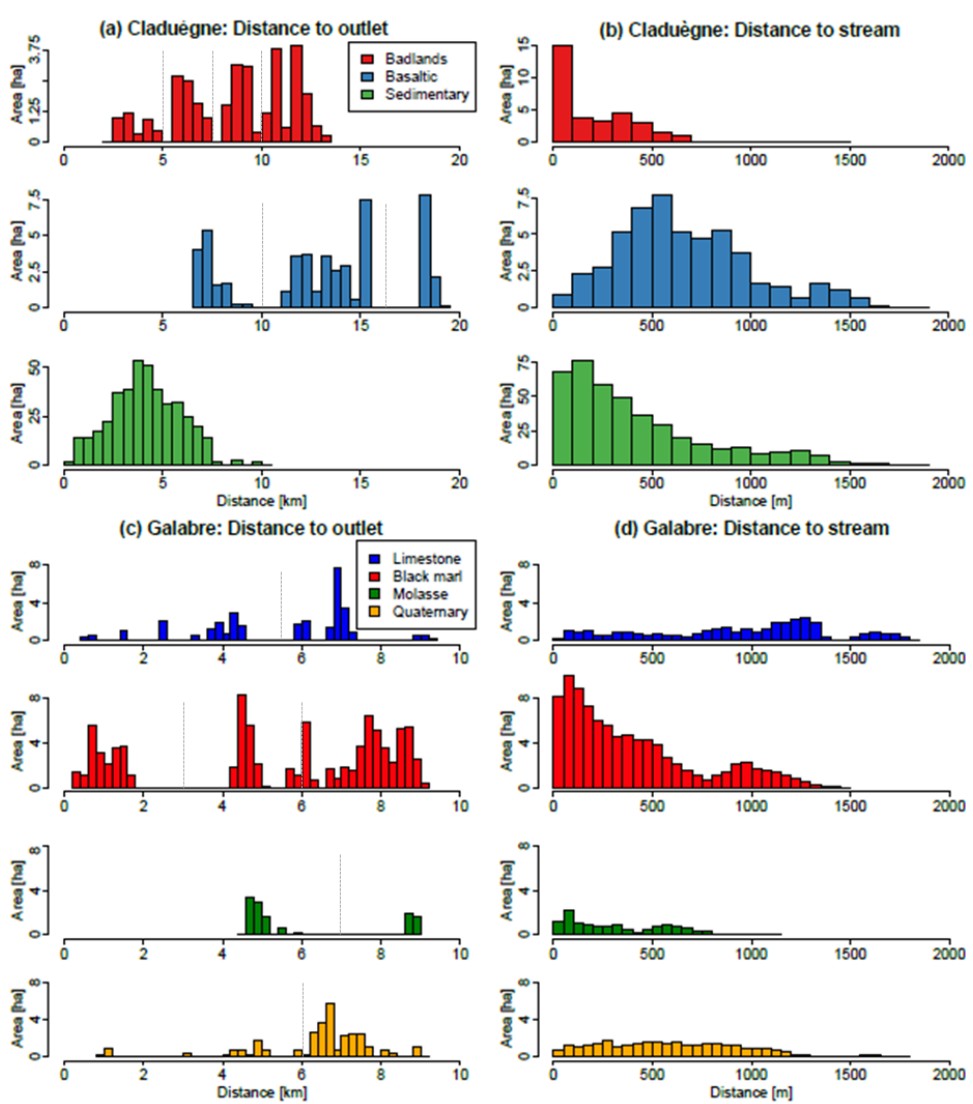

**Figure 2:** Distribution of the distance of the sources to the outlet (a for the Claduègne, c for the Galabre) and the stream (b for the Claduègne, d for the Galabre). The stream was defined with a threshold of contributing drainage area of 50 ha. The values represent distances along the flowlines that water and sediments travel following the gradient of the relief. Dashed grey lines correspond to the limits of subgroups of geological sources based on their distance to the outlet modelled in Sc 4b and 4d.



Earth **Surface**
**Dynamics**
Discussions



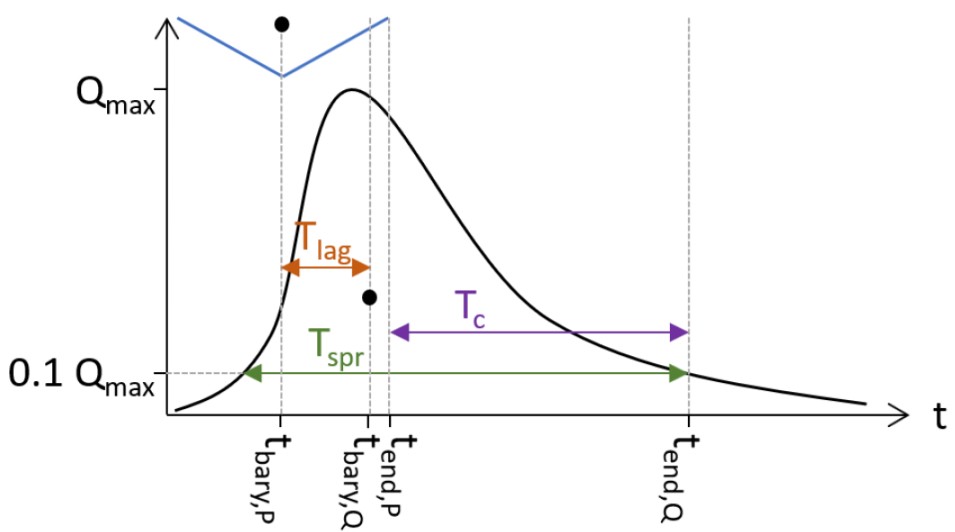



**Figure 3:** Scheme of the calculation of characteristic times $T_{lag}$, $T_c$ and $T_{spr}$ that were calculated using the simulated
liquid and solid discharges. The points represent the barycenter of the hyetograph (blue curve) and of the fraction
of discharge above the threshold of $0.1Q_{max}$ (black curve).





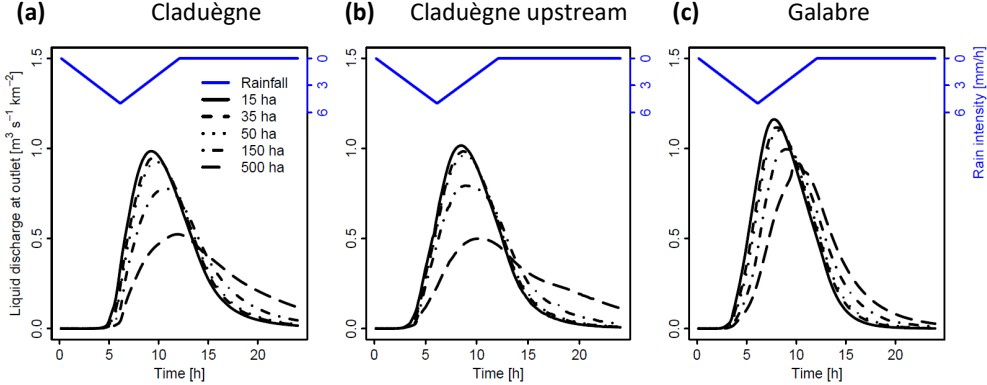

**Figure 4:** Simulated specific discharge obtained with different scenarios of model discretization at the outlet of (a) the 42km² Claduègne catchment, (b) the 20km² upstream outlet of the Claduègne where the size of the subcatchment is the same as the one of (c) the Galabre catchment. The threshold for defining the river network is varied from 15 ha to 500 ha.

Earth **Surface**
**Dynamics**
Discussions

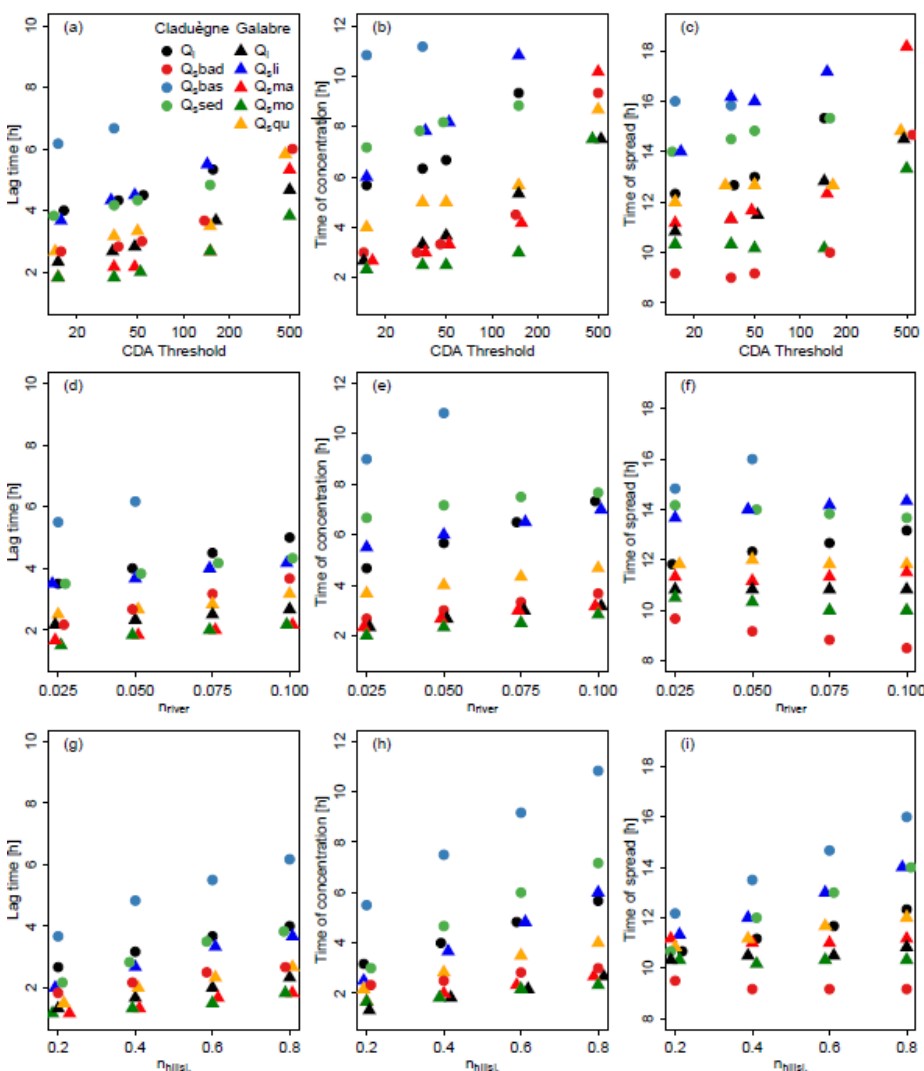

**Figure 5:** Sensitivity of lag times, times of concentration and time of spread to changing the CDA threshold (top
row), Manning's *n* in the river network (middle row) and on the hillslopes (bottom row). For each catchment the
characteristic times are given for liquid discharge (Ql) and for solid discharge (Qs) of the different source classes.
Some symbols were slightly shifted on the x-axis if they were hard to see or overlapped by other symbols.



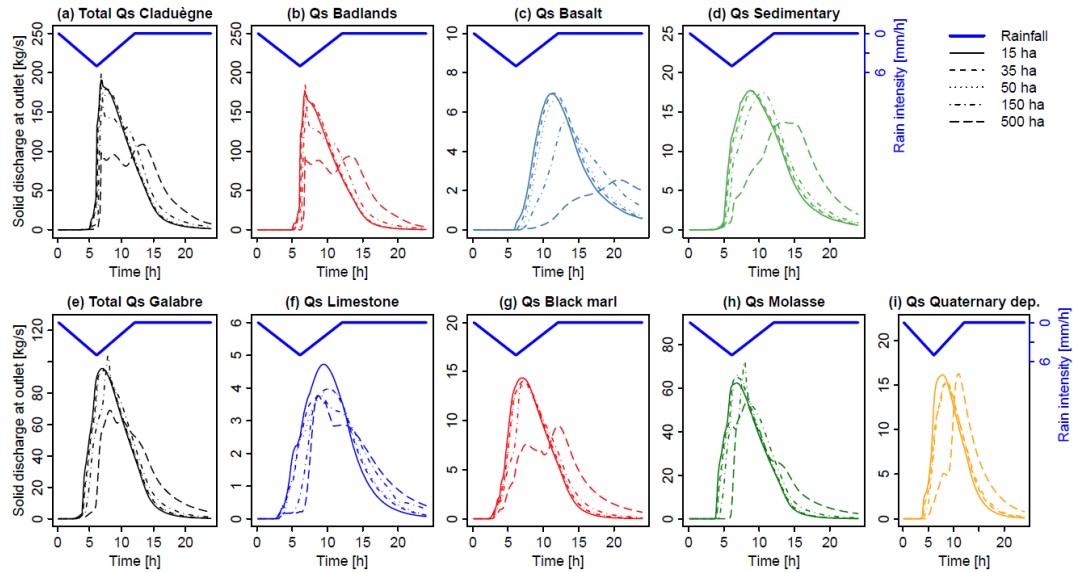


**Figure 6:** Simulated sedigraphs for total solid discharge (Qs) and for each source in the two catchments when
different values are used for the threshold of contributing drainage area (CDA) to define the river network.



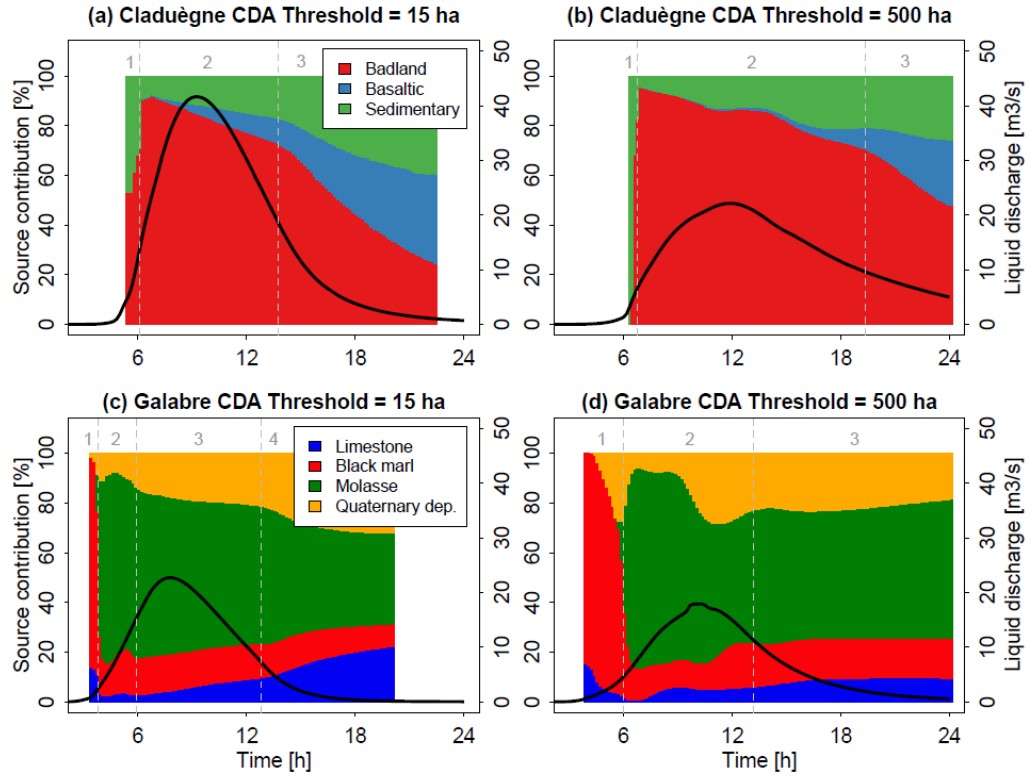


**Figure 7:** Modeled source contributions of the sediment sources in the Claduègne and Galabre catchments when
the threshold of contributing drainage area (CDA) is set to 15 ha (left, Sc. 1) or to 500 ha (right, Sc. 2d). The color
shows the contribution of the different sources to total suspended sediment load in percent. The hydrograph is
additionally shown to represent the timing of the event. The results obtained with all five CDA thresholds (15, 35,
50, 150 and 500 ha) for both catchments can be visualized in interactive figures at
https://shiny.osug.fr/app/EROSION_MODEL.2020



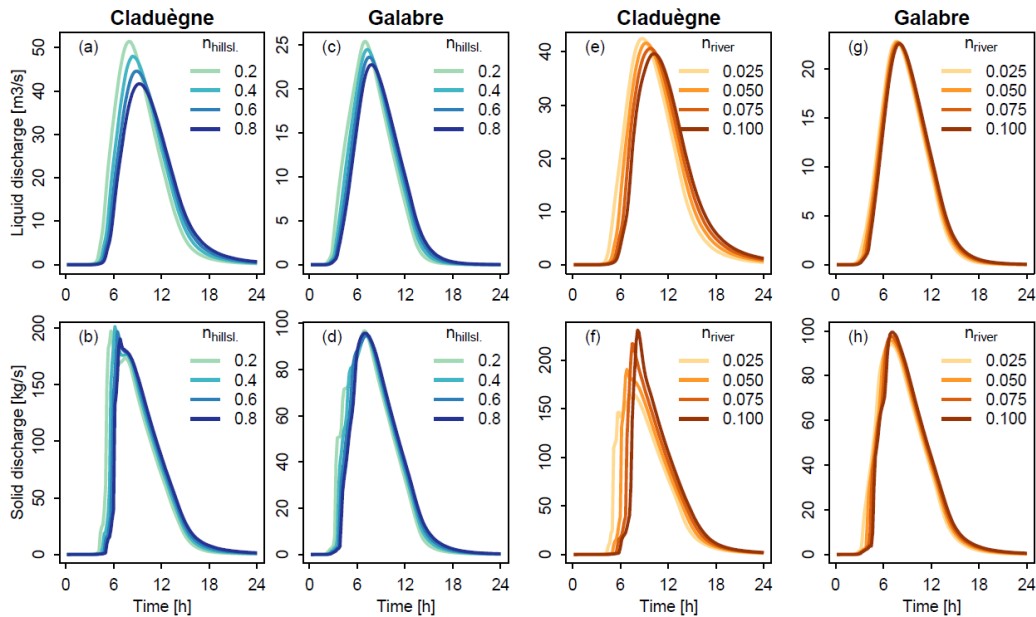

**Figure 8:** Sensitivity of modeled hydrographs (top row) and sedigraphs (bottom row) to changing Manning's roughness parameter on the hillslopes (a to d) and in the river network (e to h). For subfigures a to d $n_{river}$ was fixed to 0.05. For subfigures e to h $n_{hillsl}$ was fixed to 0.8.

Earth **Surface**
**Dynamics**
Discussions

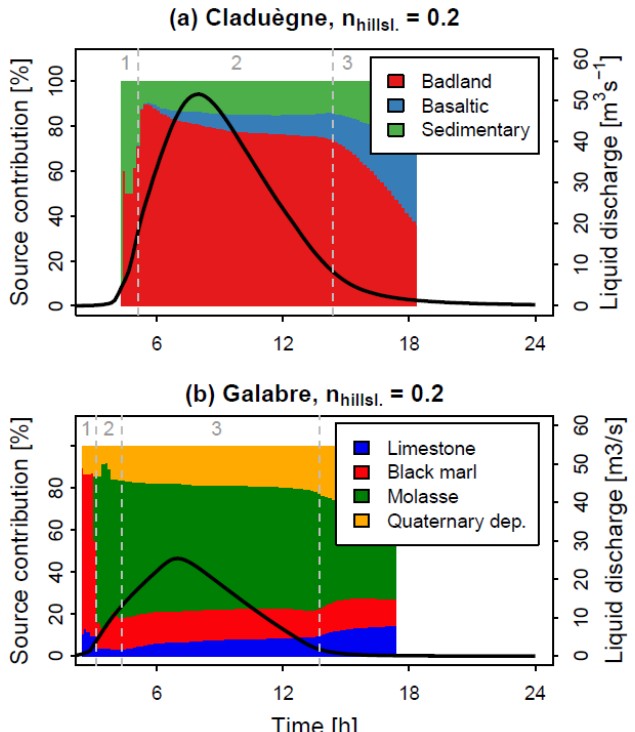

728

**Figure 9:** Modeled contributions of the sediment sources in the two catchments when Manning's *n* on the hillslopes
was set to 0.2 (Sc. 3a). The color shows the contribution of the different sources to total suspended sediment load
in percent. The hydrograph is additionally shown to represent the timing of the event. The results obtained with
all roughness values for both catchments can be visualized in interactive figures at
https://shiny.osug.fr/app/EROSION_MODEL.2020




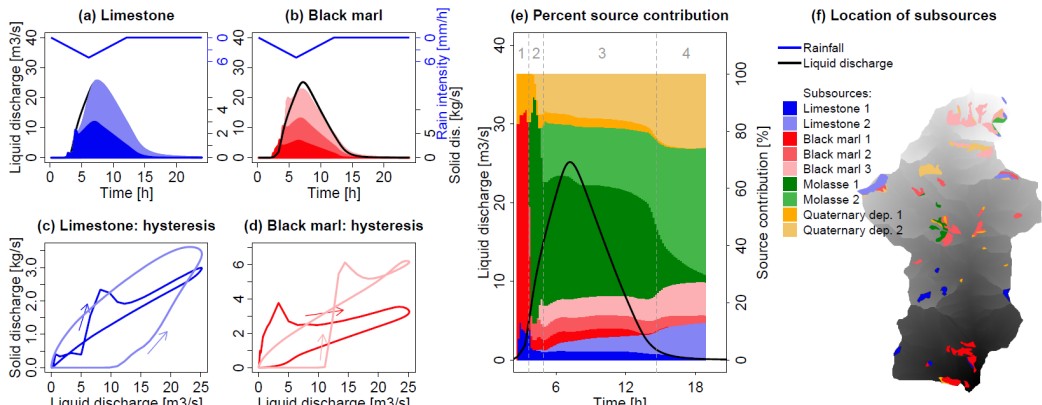



**Figure 10:** (a,b) Contribution of subsources of Limestone and Black marl that are classified according to their
distance to the outlet (Sc. 4a). The colored areas show the contribution of sources close to the outlet (darker colors)
and more distant sources (lighter colors) to the sedigraph. (c,d) shows the hysteresis loops of the subsources. (e)
shows the contribution of each subsource to total solid discharge in percent. The dashed lines and the grey numbers
above the figure distinguish different periods of the event as referred to in the text. (f) Location of the subsources
in the Galabre catchment.





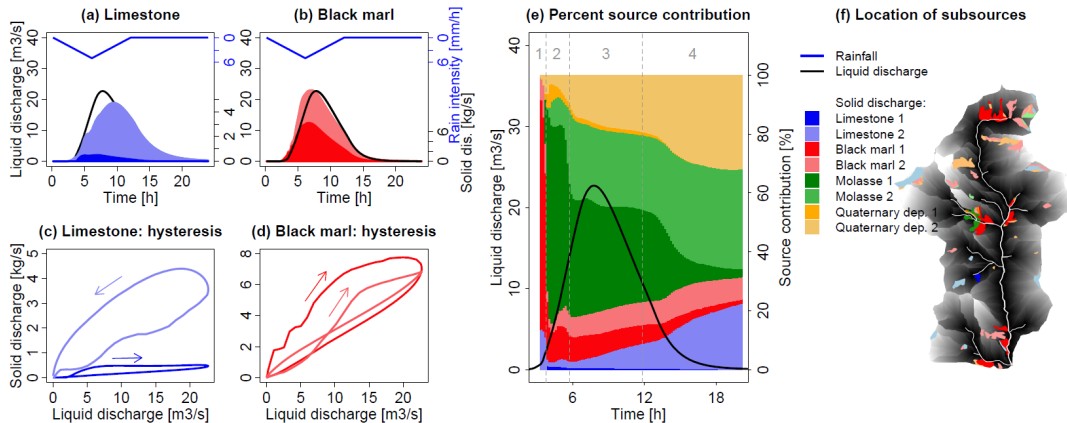

**Figure 11:** Contribution of subsources that are classified according to their distance to the stream in the Galabre catchment (Sc. 4b). For the description of the subfigures, see the caption of Figure 10.





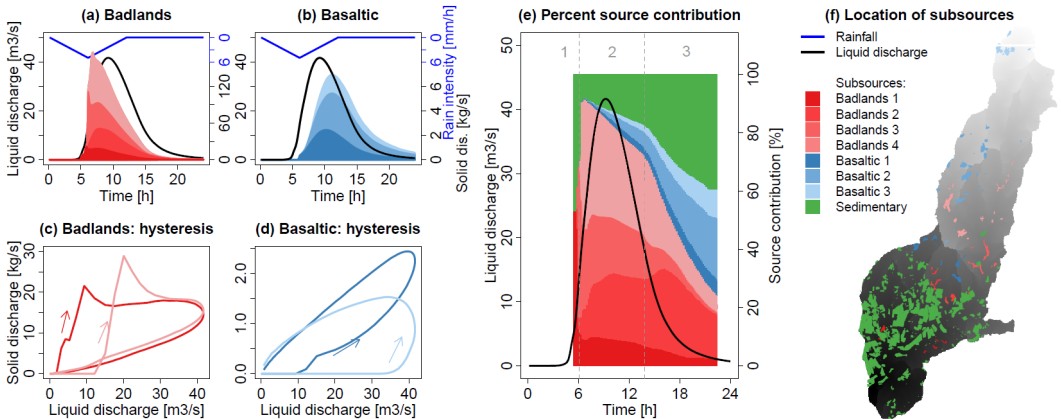

**Figure 12:** (a-b) Contribution of subsources of badlands and basaltic sources that are classified according to their distance to the outlet (Sc. 4a). The colored areas show the contribution of sources close to the outlet (darker colors) and more distant sources (lighter colors) to the sedigraph. (c-d) show the hysteresis loops of the subsources. Subfigure (e) shows the contribution of each subsource to total solid discharge in percent. The dashed lines and the grey numbers above the figure distinguish different periods of the event as referred to in the text. (f) Location of the subsources in the Claduègne catchment.



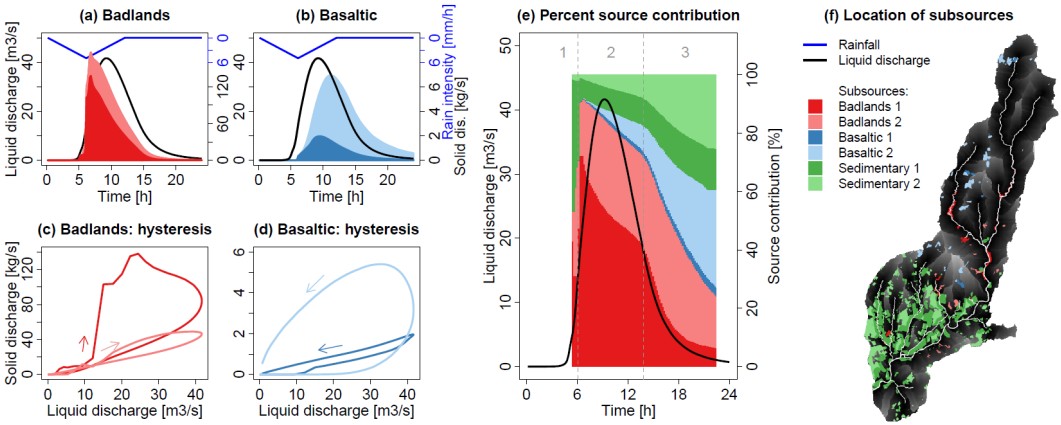



**Figure 13:** Contribution of subsources that are classified according to their distance to the stream in the Claduègne
catchment (Sc. 4b). For the description of the subfigures see the caption of Figure 12.







|  | Claduègne |  |  |  | Galabre |  |  |  |  |
|---|---|---|---|---|---|---|---|---|---|
|  | Entire catchment | Badland | Basaltic | Sedimentary | Entire catchment | Limestone | Marl | Molasse | Quaternary deposits |
| **Catchment morphology** |  |  |  |  |  |  |  |  |  |
| Area [$km^2$] | 42.24 | 0.32 | 0.52 | 4.19 | 19.55 | 0.34 | 0.93 | 0.13 | 0.33 |
| $K_G$ [-] | 1.87 | - | - | - | 1.47 | - | - | - | - |
| Slope, hillslopes | 24 ± 30 | 82 ± 68 | 11 ± 21 | 12 ± 13 | 54 ± 40 | 101 ± 127 | 67 ± 38 | 56 ± 30 | 54 ± 33 |
| Slope, river network |  |  |  |  |  |  |  |  |  |
| Intermittent streams | 6.78 | - | 9.22[a] | 6.06[a] | 19.17 | - | - | - | - |
| Main stream | 2.72 | - | 4.93[a] | 2.50[a] | 5.71 | - | - | - | - |
| **Connectivity** |  |  |  |  |  |  |  |  |  |
| Distance to outlet [$km$] | 9.18 ± 5.10 | 8.59 ± 2.82 | 12.91 ± 3.92 | 4.15 ± 1.73 | 4.75 ± 2.17 | 5.49 ± 1.99 | 5.28 ± 2.91 | 6.03 ± 1.72 | 6.25 ± 1.65 |
| Distance to stream [$km$] | 0.44 ± 0.35 | 0.21 ± 0.19 | 0.67 ± 0.34 | 0.42 ± 0.36 | 0.53 ± 0.37 | 0.89 ± 0.47 | 0.39 ± 0.35 | 0.34 ± 0.24 | 0.57 ± 0.35 |
| IC (Borselli et al., 2008) | -9.18 ± 0.61 | -8.35 ± 0.43 | -9.30 ± 0.37 | -8.75 ± 0.66 | -8.84 ± 0.75 | -7.94 ± 0.39 | -7.95 ± 0.60 | -8.19 ± 0.36 | -8.03 ± -0.42 |
| IC (Cavalli et al., 2013) | -5.85 ± 0.53 | -5.50 ± 0.34 | -6.34 ± 0.50 | -5.73 ± 0.50 | -4.56 ± 0.50 | -4.52 ± 0.33 | -4.57 ± 0.55 | -4.81 ± 0.35 | -4.56 ± 0.40 |
| **Erodibility** |  |  |  |  |  |  |  |  |  |
| Suspended sediment yield [$t\,y^{-1}$] | 15947 | 12394 | 1084 | 2469 | 12856 | 953 | 1956 | 7474 | 2473 |
| Specific yield [$t\,km^{-2}y^{-1}$] | 380 | 38623 | 2087 | 589 | 666 | 2780 | 2113 | 57075 | 7418 |
| Rain erodibility $\alpha^{b)}$ [$g\,mm^{-1}m^{-2}$] | 3.1 | 37.5 | 2.0 | 0.6 | 7.4 | 2.8 | 2.1 | 57.1 | 7.4 |

**Table 1:** Characteristics of the two catchments and the erosion zones. KG is Gravelius' compactness indicator defined as the ratio between the catchment perimeter (P) and the one of a circle with equal surface. The values given for the slopes on the hillslopes, the distance to the outlet, the distance to the streams and the two connectivity indicators (IC) represent the mean +/- standard deviation. The mean slopes in the river network are given for the entire network including intermittent streams (defined with a threshold of CDA of 15 ha) and for the main, perennial network (CDA of 500 ha). a) The values correspond to the slope in the river network on the basaltic plateau and on sedimentary geology and are not limited to the erosion zones. b) Rainfall erodibility corresponds to the mass of sediment detached on 1m² by 1mm of rain (Cea et al., 2015).





| Sc. | Th$_{CDA}$ [ha] | Source classification | n$_{river}$ [-] | n$_{hillsl.}$ [-] | Aim |
|---|---|---|---|---|---|
| 1 | 15 | Geology | 0.050 | 0.8 | Basic scenario |
| 2a | 35 | Geology | 0.050 | 0.8 | Impact of the river network threshold |
| 2b | 50 | Geology | 0.050 | 0.8 | |
| 2c | 150 | Geology | 0.050 | 0.8 | |
| 2d | 500 | Geology | 0.050 | 0.8 | |
| 3a | 15 | Geology | 0.050 | 0.2 | Impact of the parameterization of Manning's n |
| 3b | 15 | Geology | 0.050 | 0.4 | |
| 3c | 15 | Geology | 0.050 | 0.6 | |
| 3d | 15 | Geology | 0.025 | 0.8 | |
| 3e | 15 | Geology | 0.075 | 0.8 | |
| 3f | 15 | Geology | 0.100 | 0.8 | |
| 4a | 15 | Geology and distance to the outlet | 0.050 | 0.8 | Dynamics between more and less connected sources |
| 4b | 15 | Geology and distance to the stream | 0.050 | 0.8 | |
| 4c | 15 | Geology and distance to the outlet | 0.100 | 0.2 | |
| 4d | 15 | Geology and distance to the stream | 0.100 | 0.2 | |


**Table 2:** Model scenarios (Sc) detailed according to the value of the contributing drainage area threshold to define
the river network (ThCDA), the approach to classify the sources, the values for Manning's roughness parameter
(n) in the river network and on the hillslopes and the aim of the respective scenario.





|  | 1 Basic Scenario | 2a $Th_{CDA}$ = 35 ha | 2b $Th_{CDA}$ = 50 ha | 2c $Th_{CDA}$ = 150 ha | 2d $Th_{CDA}$ = 500 ha | 3a $n_{hillsl.}$ = 0.2 | 3b $n_{hillsl.}$ = 0.4 | 3c $n_{hillsl.}$ = 0.6 | 3d $n_{river}$ = 0.025 | 3e $n_{river}$ = 0.075 | 3f $n_{river}$ = 0.100 |
|---|---|---|---|---|---|---|---|---|---|---|---|
| **Claduègne** | | | | | | | | | | | |
| $T_{lag,Q_l}$ [h] | 4.00 | 4.33 | 4.50 | 5.33 | NA | 2.67 | 3.17 | 3.67 | 3.50 | 4.50 | 5.00 |
| $T_{c,Q_l}$ [h] | 5.67 | 6.33 | 6.67 | 9.33 | NA | 3.17 | 4.00 | 4.83 | 4.67 | 6.50 | 7.33 |
| $T_{spr,Q_l}$ [h] | 12.33 | 12.67 | 13.00 | 15.33 | NA | 10.67 | 11.17 | 11.67 | 11.83 | 12.67 | 13.17 |
| $Q_{l,max}$ [$m^3 s^{-1}$] | 41.65 | 40.16 | 39.14 | 32.91 | 22.14 | 51.44 | 48.00 | 44.57 | 42.51 | 40.67 | 39.64 |
| $Q_{s,max}$ [$kg\,s^{-1}$] | 191.04 | 198.67 | 183.24 | 169.41 | 108.65 | 197.45 | 201.52 | 196.98 | 163.88 | 217.06 | 230.97 |
| $T_{lag,Q_s}$ bad [h] | 2.67 | 2.83 | 3.00 | 3.67 | 6.00 | 1.83 | 2.17 | 2.50 | 2.17 | 3.17 | 3.67 |
| $T_{c,Q_s}$ bad [h] | 3.00 | 3.00 | 3.33 | 4.50 | 9.33 | 2.33 | 2.50 | 2.83 | 2.67 | 3.33 | 3.67 |
| $T_{spr,Q_s}$ bad [h] | 9.17 | 9.00 | 9.17 | 10.00 | 14.67 | 9.50 | 9.17 | 9.17 | 9.67 | 8.83 | 8.50 |
| $T_{lag,Q_s}$ bas [h] | 6.17 | 6.67 | NA | NA | NA | 3.67 | 4.83 | 5.50 | 5.50 | NA | NA |
| $T_{c,Q_s}$ bas [h] | 10.83 | 11.17 | NA | NA | NA | 5.50 | 7.50 | 9.17 | 9.00 | NA | NA |
| $T_{spr,Q_s}$ bas [h] | 16.00 | 15.83 | NA | NA | NA | 12.17 | 13.50 | 14.67 | 14.83 | NA | NA |
| $T_{lag,Q_s}$ sed [h] | 3.83 | 4.17 | 4.33 | 4.83 | NA | 2.17 | 2.83 | 3.50 | 3.50 | 4.17 | 4.33 |
| $T_{c,Q_s}$ sed [h] | 7.17 | 7.83 | 8.17 | 8.83 | NA | 3.00 | 4.67 | 6.00 | 6.67 | 7.50 | 7.67 |
| $T_{spr,Q_s}$ sed [h] | 14.00 | 14.50 | 14.83 | 15.33 | NA | 10.67 | 12.00 | 13.00 | 14.17 | 13.83 | 13.67 |
| **Galabre** | | | | | | | | | | | |
| $T_{lag,Q_l}$ [h] | 2.33 | 2.67 | 2.83 | 3.67 | 4.67 | 1.33 | 1.67 | 2.00 | 2.17 | 2.50 | 2.67 |
| $T_{c,Q_l}$ [h] | 2.67 | 3.33 | 3.67 | 5.33 | 7.50 | 1.33 | 1.83 | 2.17 | 2.33 | 3.00 | 3.17 |
| $T_{spr,Q_l}$ [h] | 10.83 | 11.33 | 11.50 | 12.83 | 14.50 | 10.33 | 10.50 | 10.50 | 10.83 | 10.83 | 10.83 |
| $Q_{l,max}$ [$m^3 s^{-1}$] | 22.71 | 21.83 | 21.50 | 19.47 | 17.89 | 25.38 | 24.43 | 23.58 | 22.79 | 22.61 | 22.54 |
| $Q_{s,max}$ [$kg\,s^{-1}$] | 95.70 | 94.73 | 94.29 | 103.65 | 69.15 | 96.64 | 95.15 | 94.54 | 94.08 | 97.66 | 99.52 |
| $T_{lag,Q_s}$ li [h] | 3.67 | 4.33 | 4.50 | 5.50 | NA | 2.00 | 2.67 | 3.33 | 3.50 | 4.00 | 4.17 |
| $T_{c,Q_s}$ li [h] | 6.00 | 7.83 | 8.17 | 10.83 | NA | 2.50 | 3.67 | 4.83 | 5.50 | 6.50 | 7.00 |
| $T_{spr,Q_s}$ li [h] | 14.00 | 16.17 | 16.00 | 17.17 | NA | 11.33 | 12.00 | 13.00 | 13.67 | 14.17 | 14.33 |
| $T_{lag,Q_s}$ ma [h] | 1.83 | 2.17 | 2.17 | 2.67 | 5.33 | 1.17 | 1.33 | 1.67 | 1.67 | 2.00 | 2.17 |
| $T_{c,Q_s}$ ma [h] | 2.67 | 3.00 | 3.33 | 4.17 | 10.17 | 1.67 | 2.00 | 2.33 | 2.33 | 3.00 | 3.17 |
| $T_{spr,Q_s}$ ma [h] | 11.17 | 11.33 | 11.67 | 12.33 | 18.17 | 11.17 | 11.00 | 11.00 | 11.33 | 11.33 | 11.50 |
| $T_{lag,Q_s}$ mo [h] | 1.83 | 1.83 | 2.00 | 2.67 | 3.83 | 1.17 | 1.33 | 1.50 | 1.50 | 2.00 | 2.17 |
| $T_{c,Q_s}$ mo [h] | 2.33 | 2.50 | 2.50 | 3.00 | 7.50 | 1.67 | 1.83 | 2.17 | 2.00 | 2.50 | 2.83 |
| $T_{spr,Q_s}$ mo [h] | 10.33 | 10.33 | 10.17 | 10.17 | 13.33 | 10.33 | 10.17 | 10.33 | 10.50 | 10.00 | 10.00 |
| $T_{lag,Q_s}$ qu [h] | 2.67 | 3.17 | 3.33 | 3.50 | 5.83 | 1.50 | 2.00 | 2.33 | 2.50 | 2.83 | 3.17 |
| $T_{c,Q_s}$ qu [h] | 4.00 | 5.00 | 5.00 | 5.67 | 8.67 | 2.17 | 2.83 | 3.50 | 3.67 | 4.33 | 4.67 |
| $T_{spr,Q_s}$ qu [h] | 12.00 | 12.67 | 12.67 | 12.67 | 14.83 | 10.83 | 11.17 | 11.67 | 11.83 | 11.83 | 11.83 |

Change [%]    0-9   10 - 19   20 - 29   30 - 49   50 - 69   70 - 89   90 - 119   120 - 149   150 - 179   ≥ 180

**Table 3:** Calculated characteristics of modeled hydrographs and sedigraphs for the different scenarios. Abbreviations: $T_{lag;Ql}$: lag time of liquid discharge, $T_{c;Ql}$: time of concentration of liquid discharge, $T_{spr;Ql}$: spread of the hydrograph, $Q_{l;max}$: peak of liquid discharge. $Qs$ refers to solid discharge and the characteristic times are calculated for each source separately (i.e. badlands, basaltic and sedimentary in the Claduègne catchment; limestone, black marl, molasses and quaternary deposits in the Galabre catchment). The background color of the cells represents the percent change of each value with respect to the basic scenario.