# Peer review of "How do modeling choices impact the representation of"

_Earth Surface Dynamics, 2020_

## Referee Comment (RC1) · Anonymous Referee #1 · 26 Oct 2020

**1   Summary**

Uber et al. present a numerical modeling study that explores how modeling choices related to computational mesh generation, parameterization, and source-classification grouping influences a variety of output metrics describing hydrograph and sedigraph characteristics. The authors focus their work in two well studied mesoscale catchments and connect the results of their sensitivity analysis with basin-scale characteristics (e.g., mean slope).

[Figure]

The presented research represents a valuable contribution to understanding how the choices made in setting up a computational model influence model results. Placing this level of attention on model set up is rarely described in formal publications, yet is critical to understanding when and how models can be applied to understand and/or predict systems of interest. In addition, the author's well designed numerical experiments permits an assessment of how discretization and parameterization impact basin hydrograph and sedigraph dynamics.

Below I describe comments and recommendations first in narrative form and then as line-level comments. My most substantial concern is that the paper lacks an overarching introduction to the study design—a section in which the authors set up the specific questions or hypothesis that they seek to address and connect them with a conceptual description of their numerical experiment design. A related comment is that I found the explanation of the modeling choices difficult to follow. Both of these issues meant that it was difficult to connect the study design and methods with the results and discussion.

I recommend acceptance after major revisions and look forward to seeing this paper published.

**2 Narrative Comments**

**2.1 Addition of an "Study Design" Section**

The experimental design employed by the authors is valid and appropriate for the questions that they seek to pose. However, I found description clearly connecting the big picture questions ("what controls sediment flux from mesoscale watersheds") to the scenario design currently introduced by Section 3.4 and Table 2 was missing, or spread across too many sections of the paper.

I recommend that a new section be placed immediately after the introduction. In this section you would describe your experimental design and connect it to the big picture you have laid out in your introduction. Such a section would include the specific questions and hypotheses each scenario's experiment seeks to answer and an explanation of why this question was targeted.

While the reader may not know the details of the two sites or the model, your introduction should provide enough information such that this section can come before the more detailed methods section. Such a section will introduce to the reader the concrete questions your scenarios were designed to address.

Such as section should a description of the type of model analysis method used (e.g., a series of one-at-a-time sensitivity studies) and explain why this sort of method is appropriate to address the study objectives. Pianosi et al. (2016) is a good place to start for background on this topic. This will help the reader understand the type of results you will obtain.

In such a section, I would also like to see an introduction to why two catchments are used and why calculating whole-catchment connectivity metrics (described in Section 3.1); e.g., doing the same set of simulations across two catchments with different geology/land use/etc allows you to isolate how transferable your results are to catchments with different properties. This would also allow you to set up why you calculate a variety of catchment connectivity metrics (presented in Table 1) and explicitly state that you will eventually work to connect those connectivity metrics with the variability identified by the sensitivity analysis (a start at this is done at L461).

**2.2   Improve explanation of modeling choices**

The core of the study hinges on connecting the modeling set up described in Section 3.3 to the scenarios described in Section 3.4. However, I found it difficult to connect these two sections, mostly because I found it hard to follow exactly what the authors varied in their modeling set up.

The most constructive form of feedback I think I can provide here is a summary of what I understood after reading the paper four times, as well as what I would recommend so that I might have understood this after the first reading.

Based on my reading, what I understand is that there Iber requires a computational mesh, and the mesh size can vary in space. Each mesh cell has a value for Manning's $n$ and a value for $\alpha$. *Choice 1*: The considered area is divided up into three conceptual domains which influence the grid cell size and Manning's $n$ value based on the CDA (hillslope, channel, badlands). Based on the delineation of these domains the mesh is discretized.

Next the mesh is parameterized with a spatially variable for Manning's $n$ value. You might have chosen to let Manning's $n$ vary smoothly, or something else, but you have chosen that the domain will get two Manning's values (channel and hillslope). *Choice 2* focuses on those values.

While water can fall on and run across the entire computational mesh, sediment can only be sourced from the bare bedrock areas. In these areas, the propensity to produce sediment is parameterized by $\alpha$.

I don't think the following was ever stated, but in order to produce the source proportion sedigraphs, I believe that some method of source tracking can be chosen in order to elucidate the dynamics of the basin. Different classification of these tracked sources is represented by *Choice 3* (I think).

Thus Scenarios 2a–2d focus on Choice 1, Scenarios 3a–f focus on Choice 2, and I think that different delineations of source tracking (Choice 3), along with different choices for Manning's $n$ yield Scenario 4.

I would recommend the following to the authors:

- Revise section 3.3 to describe more clearly what the modeling choices are such that they set the reader up to understand the details of scenario design discussed in the following section.

- In Section 3.3 or in the new "study design" section proposed above, explain why these choices are important to focus on. Are they the only choices? Are they the only ones which carry uncertainty? There are many things you might have focused on (e.g., assess the sensitivity to the channel grid cell size), but you chose these elements, why? To be clear, I think the elements you've chosen are great, I just want more description of why they were chosen.

- Clarify how the source classification is represented in model specification. Does this choice not influence the model physics, but just the model output that permits a different view on the dynamics?

- Explain why sediment is only sourced from the bare bedrock.

2.3   Improve connection between study design and discussion

The structure of the discussion roughly follows the three non-base case scenarios and presents the most salient aspects of the results. However, within each of the major discussion sections, I found the text difficult to follow. I suspect that by being more explicit about the target questions and hypotheses earlier in the text the authors will be able to very lightly restructure the discussion such that the reader is easily able to connect the discussion with the study intent and numerical experiments.

In addition, the end of the discussion starts to tie together the basin-scale metrics presented in Table 1 and the numerical modeling results. It would be beneficial to introduce earlier on that you will do this and describe in more detail how this is accomplished (e.g., regression, rank correlation). Knowing that this sort of analysis is coming will help explain why all of the basin-scale metric are calculated and discussed starting at L136.

**2.4 Figures**

The interactive figures provided by Uber et al. (2020) are a fantastic complement to the paper. I might consider adding catchment as a facet (e.g., facet grid with scenario catchment) because this would facilitate comparison between catchments. I'd also like to applaud your consistency in the use of color to denote geological unit across figures. This should be a standard expectation, but it isn't, and it makes comprehension much better.

My primary concern with figures relates to the maps presented in Figure 1. This figure shows us inconsistent information across the two catchments (e.g., badlands only shown in 1a) and does not show us all of the information used in the modeling study that is the focus of the work. I recommend that Figure 1 be redrafted into a series of rows that shows the reader the main elements used in model initialization for each catchment. For example, row one might show a shaded relief map with the river system and badlands areas, row 2 would show the considered geologic units used, row 3 might show the weighting factor $W$ presented by Borselli et al. (2008), while row 4 would show the roughness based weighting factor of Cavalli et al. (2013).

**2.5 Code availability**

For the purposes of computational reproducibility, state the version of Iber used.

No statement has been made about model input file availability. Such files should be digitally archived for the purpose of reproducibility.

**3   Line Level Comments**

Bullet points in this Section indicate "<LineNumber>", "F<Figure Number>", or
"T<Table Number>".

The term "Mediterranean and mountainous" is used a few times, first here.
Mediterranean could be interpreted a few ways: e.g., places with a Mediterranean
climate, places near the Mediterranean. Recommend being more specific about
what is meant.

Recommend giving an example of your objectives and thus how structural con-
nectivity is represented to anchor this abstract concept on a concrete example or
two.

I suspect the sentence that ends in this line needs a reference.

Be more specific about which models and provide examples with associated ref-
erences.

Additional subsubsection headers would have helped me understand this section
more easily. For example Section 3.1 discusses both a description of the catch-
ments and connectivity metrics calculated, and Section 3.3 discusses many dif-
ferent aspects of the model set up. I would split each of these subsections into
multiple subsubsections.

A few lines or a paragraph summarizing the similarities and differences of the two
catchments would benefit the reader here.

Some statements about why these connectivity metrics were chosen would ben-
efit the reader. In addition, explain (here or in something like the proposed "Study
Design" section) what you expect to learn from these metrics and how they are
used.

The distance to the outlet metric has been called the "width function" by the land-
scape evolution modeling community Hancock et al. (2010, 2002). Work by this
community has shown that it is not a particularly good metric for comparing catch-
ment topography, but is a does provide a good assessment of hydrology. It may
be useful to connect with this literature.

Mathematically represent the connectivity indices of Borselli et al. (2008) and
Cavalli et al. (2013) here so that the reader can more clearly understand what
they represent.

171–173 This detail of model set up should be located elsewhere. Probably is a subsub-
section of Subsection 3.3 (see also the comment at L237 and 289.

Being able to connect this discussion of badlands in model set up to a consistent
picture of where badlands are located is why I mentioned earlier that Figure 1
should be revised to include consistent information about each catchment.

Connect and justify the choice of a 5 m minimum grid size with relevant field
observations and the numerics of the Iber model? E.g., how does this compare
with the range of values for channel width in each catchment? Do the numerics
of Iber benefit from a relationship between minimum grid cell size and channel
width (e.g., smallest grid cell = channel width, 10 grid cells = channel width).

20 m seems like a rather large grid cell size for gullied areas. Explain and/or
justify this value.

The erosion source locations should be shown in Figure 1 in addition to the sub-
plots shown in later figures.

If I'm interpreting this correctly, I believe you are saying that sediment production
can only occur in the areas of bare bedrock. This should be explained further
and justified. In addition, discuss how this model set up decision impacts the implications of this study for overall soil erosion (as these bare bedrock patches only make up a small portion of the study watershed).

227–236 It is difficult to understand if this section of text is summarizing the work of Uber et al. (2019) or if it is presenting an analysis of modeling results. Revise to clarify this point.

Introduce the units of $\alpha$ when the variable is first presented.

No discussion of time discretization, model run duration, or external forcing (e.g., rain) is present in the prior subsection. These elements of model set and running should be discussed.

Based on the results presented, it appears that Iber has the capability of tracking the source of water/sediment as it moves through the catchment and that how these source regions are grouped is what is meant by the "source classification" column of Table 2. This aspect of the model should be discussed. As best as I can tell this is a critical aspect of interpreting Scenario 4.

In addition, it is not clear whether this choice of model set up impacts the dynamics of water and sediment (or if it just impacts how they are analyzed). E.g., are simulation 1 and 4a and 4b the same simulation just analyzed/post processed differently?

The simulations of Scenario 3 represent two one-at-a-time sensitivity studies (Sc. 3a–3c for sensitivity to hillslope Manning's $n$ and Sc. 3d–3f for channel). Recommend using more formal language to describe the numerical experiments as it will help the reader anticipate the type of results presented.

It is not clear to me how the the different options for source classification of Scenario 4 relate to changes in the parameterization of the model. Were different values of $\alpha$ used? Something else? Clarify.

In addition, these scenarios include two options for the Manning's $n$ value, the base case and one in which the hillslope value is low and the channel value is high. The results of Scenarios 4c and d are discussed at L454. Formally introduce what the purpose of this sub-scenario is.

272–274 This sentence, in which you link the changes to the model set up with a hypothesis is exactly the sort of text that a "Study Design" section would benefit from. Recommend that similar sentences for each scenario exist and be present in such a section.

This section clearly describes what model output metrics were used, however it does not explain why these output metrics were chosen or justify why the are appropriate given the overall goals of the study. This section should be expanded to include this information.

This sentence describing model run details should go elsewhere in the text. Probably in a section on external forcing, along with the text currently located at L171–173 (see comment at L237).

Be more specific about which aspects of the model. Some aspects are sensitive and some are not.

Connect this statement with new text earlier in the paper describing why two catchments are used. Set the reader up for this sort of discussion by explaining why two catchments are used, and comparing/contrasting them.

Justify why this is a reasonable interpretation and connect with literature.

This statement presents a different conclusion than Table 3 and the text near L296 which states that different CDA values result in output metric variability. These three elements of results and discussion should be consistent.

344-350 The purpose and reasoning of the argument you advance here is not clear. As you highlight it in the conclusion (L487) I believe you think it is an important point. Recommend this text be revised.

The section of Table 2 that shows the results of Scenario 3 indicates that changing Manning's $n$ in the hillslope has a larger impact on the results than changing the channel value. This should be discussed.

What is meant by "more stable"?

Here and elsewhere, sensitivity should be presented as a relative measure. E.g., this output was more sensitive to choice/parameter A than to choice/parameter B. Without the comparison the statement is uninterpretable.

Here you discuss both a contrast between the two catchments, the analysis of Scenario 4, and connecting basin-wide metrics of IC with the sensitivity results. Recommend structuring the section to help the reader anticipate this.

Introduce this idea in the study design.

397–399 This has already been stated.

Add a figure reference.

More specific. E.g., close = first, or something different?

It is not clear if Scenario 4 represents a different approach to tracking something else? Because the description of how Sc. 4 was constructed is incomplete is is nearly impossible to understand the results of Sc. 4.

423–425 Give the reader a little more context about "typical interpretations of discharge-sediment flux hysteresis" and provide a description of what a clockwise vs counterclockwise loop means.

Not sure what is meant by this sentence.

Unclear if distance to the outlet (or stream) being considered is related to the parameterization or the analysis of the results.

This sentence starts a new line of inquiry: which basin-wide metrics (Table 1) best predict the sensitivities documented by the numerical experiments. A more explicit discussion of the methods used here (e.g., comparing basin wide metrics to sensitivity ranking) should be added to the methods. In addition, the description of this analysis should be expanded.

This sentence is not clear.

It is not clear that your study design supports this type of analysis. To my ability to tell you have not varied the location and/or erodibility of the sediment sources within the catchment. As such, your study design does not permit assessment of how variability in location of sediment sources influences the output metrics.

The point you are making here is not clear, mostly because the text introduced at L344-350 is not clear.

Unclear how the study is about source soils when the only erodible material is the exposed bedrock. This should be addressed here and earlier in the text.

ost Figures In the many multi-panel plots I recommend use of consistent x and y axis limits and/or explicit notation of inconsistent axis limits in Figure captions.

F10–13 The panel (f) is the sort of information that would be great to have in a revised Figure 1. The background color scheme for the inset maps (distance to outlet, distance to stream) should be represented by a legend.

T2 The layout of the table makes it difficult to see the difference between the scenario 4 options.

T3   1. Why are the simulations used for Scenario 4 not in the table?

2. Recommend adding some vertical lines to help guide the viewer in separating Sc. 1, Sc. 2, and the two halves of Sc. 3.

3. Overlaying the table text on top of a tile plot is a great addition. However, the darkest blue values make reading the text impossible.

4. Not clear why some values have NA, explain.

**References**

Borselli, L., Cassi, P., and Torri, D.: Prolegomena to sediment and flow connectivity in the landscape: a GIS and field numerical assessment, Catena, 75, 268–277, 2008.

Cavalli, M., Trevisani, S., Comiti, F., and Marchi, L.: Geomorphometric assessment of spatial sediment connectivity in small Alpine catchments, Geomorphology, 188, 31–41, 2013.

Hancock, G., Willgoose, G., and Evans, K.: Testing of the SIBERIA landscape evolution model using the Tin Camp Creek, Northern Territory, Australia, field catchment, Earth Surface Processes and Landforms: The Journal of the British Geomorphological Research Group, 27, 125–143, 2002.

Hancock, G., Lowry, J., Coulthard, T., Evans, K., and Moliere, D.: A catchment scale evaluation of the SIBERIA and CAESAR landscape evolution models, Earth Surface Processes and Landforms, 35, 863–875, 2010.

Pianosi, F., Beven, K., Freer, J., Hall, J. W., Rougier, J., Stephenson, D. B., and Wagener, T.: Sensitivity analysis of environmental models: A systematic review with practical workflow, Environmental Modelling & Software, 79, 214–232, 2016.

Uber, M., Legout, C., Nord, G., Crouzet, C., Demory, F., and Poulenard, J.: Comparing alternative tracing measurements and mixing models to fingerprint suspended sediment sources in a mesoscale Mediterranean catchment, Journal of Soils and Sediments, 19, 3255–3273, 2019.

Uber, M., Nord, G., Legout, C., and Cea, L.: Modeled contributions of sediment sources to total suspended sediment flux in two mesoscale catchments, https://doi.org/http://dx.doi.org/10.17178/EROSION_MODEL.2020, 2020.

---

## Editor Comment (EC1) · Greg Hancock (Editor) · 30 Nov 2020

Review of 'How do modelling choices impact the representation of structural connectivity and the dynamics of suspended sediment fluxes in distributed soil erosion models' by Uber et al. This is a timely paper. Given the number of hydrology and sediment transport models available understanding the sensitivity of parameters is extremely important. Therefore, the topic is of high interest. The paper reports on an assessment of model sensitivity in two catchment in France. The field data and numerical experiment is nicely done. However, there a few comments that need to be addressed that

can make the paper stronger. The Abstract summarises the paper nicely. However, the Introduction needs some attention. At the end of the Introduction, I largely agree and understand all the you have described, but I am not sure where the paper is really going. I have read the Introduction several times and it is not clear what you are really going to do. This leads to a comment about Section 3.4 (and its logic) which is somewhat difficult to rationalise in terms of the various model runs and setup. The Introduction needs to be refocussed with a much stronger and defined aim particularly at the end of the section. The sentence on lines 72-74 seems to summarise the overall intent of the paper. While the sentences on lines 92-94 are quite vague. Line 174- Soil erosion module I have no problem with using a single layer in an instance like this. However, the model used here only models erosion? No deposition? I realise that the inclusion of deposition adds complexity and would likely slow model run time but what is the effect of neglecting this on the findings? Landscape Evolution Models have demonstrated that including deposition has a significant influence on erosion particularly gullying. I say this as you mention gullies in the Badlands in Section 3.3. A further issue is that you are only modelling suspended sediment? Is this the case? What about bedload? Is the quantity of bedload significant? Should you be examining total load? Line 420-424. Here you talk about total solids. Does this include bedload? Or is it suspended load? Conclusion. Can this be rewritten to summarise succinctly the interesting work here. A Conclusion should summarise and largely be standalone with data presented. I suggest that lines 489-492 have been discussed elsewhere. As presented it reads like an extension of the Discussion and does not do the paper justice. Other issues Line 128. What is 'molasses'? I really liked the interactive figures

---

## Short Comment (SC1) · 5 Dec 2020

The paper by Uber et al. presents an interesting analysis of sediment load variability by catchment-scale physically-based distributed PBD rainfall-runoff modelling. The authors demonstrate that the "actual location of sediment sources was more important than the choices made during discretization and parameterization of the model", where the analyzed modelling choices were the impact of contributing drainage area and Manning's roughness. Indeed, I agree that the location of major sediment sources is key for understanding sediment flux variability in any catchment, and that PBD mod-

elling (such as that presented in this work) is an excellent way to represent the dynamic connectivity of catchments for sediment flux and better understand its sensitivities.

The Uber et al. paper closes with the wish that "further studies should focus on the influence of rainfall dynamics on modelled sediment fluxes in mesoscale catchments." (line 474). In fact, we conducted such an analysis and presented it in this journal recently (Battista et al., 2020a). Based on this work and other ongoing efforts I would like to share my perspective on three broad questions on catchment-scale combined hydrology-sediment modelling that came to my mind after reading Uber et al. I believe these are relevant challenges which we need to address and tackle in the future, and I would like to hear the authors (and others) opinion.

**1. Structural or functional connectivity by hydrology-sediment modelling?**

The notion of structural connectivity is useful to describe the fact that the landscape surface (watershed) is a collection of potential sediment sources and sinks connected by topographic pathways of sediment transport. Functional connectivity is the dynamic driver, where each individual event activates different sediment sources (and sinks) depending on where and when there is surface overland flow (produced by rain or snowmelt). PBD models in reality do (or should) represent both structural and functional connectivity.

The work of Uber et al. focuses only on structural connectivity – their PBD hydro-dynamic surface flow solver is forced by a single (triangular) storm where rainfall is representing effective rainfall after infiltration (all rainfall runs off), the local erosion rate (in production areas) is a function of rain intensity, and all sediment is transported in suspension downstream (no deposition is allowed). Furthermore, rainfall is uniform in space, so only spatial signals coming from topography affect overland and channel flow. This effect together with mapped sediment sources allows them to quantify the contributions of different source areas.

In our work we instead focused on functional connectivity – our PBD hydrologicalsediment flow solver is forced by a continuous time series of hourly climate (rainfall, cloud cover, temperature), and overland flow is produced locally by exfiltration if the soil is saturated or rainfall intensity when it exceeds infiltration capacity at the surface. Sediment can be produced at different rates from production areas. The variability in sediment fluxes comes from a combination of topographic (structural) and hydrological (functional) connectivity and is the result of integrating over many storms (Battista et al., 2020a). At the hillslope scale both models have the same spatial resolution (100m) in these applications, so in this sense they are comparable. Ours is however an application to a much larger catchment.

These two modelling studies lead me to ask: Is it necessary to understand structural connectivity separately from functional connectivity? What do we gain by this, as real basins never experience the kind of hypothetical climatic driving conditions studied by Uber et al., and runoff production in reality is heavily dependent on soil moisture that varies strongly in space and time? For example in Battista et al. (2002a) we showed that rainfall spatial variability had a significant effect on sediment load by increasing sediment production rates (increasing functional connectivity by locally high runoff production), while variability in surface erodibility had the opposite effect (decreasing functional-structural connectivity by magnifying sediment buffers close to streams). We concluded that it was futile to try to quantify the structural effects separately from that of functional effects, because they clearly act together in producing the sediment flux variability at the outlet. I am not convinced that event scale analyses and explorations of structural connectivity are helping us understand the processes better, unless we understand (and are able to model) why every event has a different hydrological, i.e. overland flow and therefore erosion, response across a catchment.

**2. How do we validate the hydrology-sediment models we use?**

Nevertheless, event based analysis did teach us something. In the same basin and with the same PBD model we studied the effects of moving storm events on flood peaks (Paschalis et al., 2014). This numerical experiment showed that event flood peaks (and

thus high sediment transport rates) were affected more by temporal variability of rainfall at high resolution (within storm) than by spatial variability. But most importantly the soil moisture state at the onset of the events played a paramount role – because high soil moisture promoted clustering of saturated areas within a catchment leading to locally high overland flow production (and therefore erosion).

In the study of Battista et al. (2020a), and the paper by Uber et al., it is overland flow on hillslopes that erodes the soil and produces sediment to the fluvial network. This furthermore has to happen along overland flow paths that are continuous, otherwise sediment is deposited (not in Uber et al.) and does not reach the channels. In other words, any PBD model prediction of sediment fluxes at a catchment outlet relies on getting overland flow in the model right. In most cases it is not channel transport that matters.

This raises the questions: Are we sure we are predicting overland flow in our catchment scale models correctly? That means, does overland flow occur in the right place and at the right time during rainfall (snowmelt) events? The co-authors of Uber et al. have previously presented excellent work on such validation of PBD models at smaller scales with laboratory experiments (Cea et al.,2014), but at larger catchment scales this question of validating overland flow in PBD models remains open and deserving of attention. This may also affect the conclusions of Uber et al. (and any other modelling study of course). Any ideas in this direction would be very welcome.

**3. Identifying and tracking sediment sources?**

The highlight of the work of Uber et al. in my opinion is the simulation of the contributions of the different sediment sources to the outlets and the demonstration and discussion of the possible effects these have on the hysteretic loops of sediment versus water discharges in their two catchments. These results allow the authors to state that the actual location of sediment sources were more important than the modelling choices on the contributing drainage area and Manning's roughness. This is an important conclusion.

Combining the notions of functional hydrological-sediment connectivity raised in point 1, with the concern for correct simulation of local overland flow in point 2, brings me to this final challenge of using PBD hydrology-sediment models for the tracking of sediment origin. The results reported in Uber et al. give us some confidence that PBD models can be used for this purpose insofar some distinct sediment properties (mineralogy, geochemistry) of the source areas can be used as fingerprints. This is in my opinion a very worthwhile combination of hydrological modelling and geomorphological field data.

Using again our mesoscale study basin as an example, we have done this exercise with our PBD hydrology-sediment model in continuous simulation (13 yrs) at hourly timescales with two relevant conclusions (Battista et al., 2020b): (a) High peaks of suspended sediment concentrations observed at the catchment outlet could only be reproduced when we included an accurate geomorphological map of local sediment sources (landslides and incision reaches). Hillslope erosion by overland flow alone was not able to generate these high concentrations in our model. (b) Tracking sediment provenance in the model (similar to what Uber et al. do) with CRN 10Be isotopes showed us that our catchment can shift between channel-process and hillslope-process dominant behaviour in time depending on how we include our understanding of the local sediment production processes and rates into the model parameterization, and of course on the hydrological forcing. This means that the event scale partitioning of sediment flux to sources that Uber et al. find is indeed very relevant, but also time and parameterization dependent over longer timescales.

The integration over many events and longer hydrological forcing periods is in fact necessary to really identify individual sources (considering their production rate limitations) as main contributors to sediment flux at catchment scales. The use of sediment tracing techniques to validate PBD models in this regard, and in turn also to provide assistance to geomorphological interpretations of sediment provenance will be very helpful.

In conclusion, my comment serves to share some perspectives on the challenges we have had in combining PBD hydrology-sediment models in support of the work presented in Uber et al. It is my opinion there is much to gain from hydrological modelling combined with geomorphological process expertise in solving the sediment source-to-sink problem in a quantitative way, and the Uber et al. work is an interesting step in that direction.

**References**

Battista, G., Molnar, P., and Burlando, P.: Modelling impacts of spatially variable erosion drivers on suspended sediment dynamics, Earth Surf. Dynam., 8, 619–635, https://doi.org/10.5194/esurf-8-619-2020, 2020a.

Battista, G., Schlunegger, F., Burlando, P., and Molnar, P.: Modelling localized sources of sediment in mountain catchments for provenance studies. Earth Surf. Process. Landforms, https://doi.org/10.1002/esp.4979, 2020b.

Cea, L., Legout, C., Darboux, F., Esteves, M., and Nord, G. :Experimental validation of a 2D overland flow model using high resolution water depth and velocity data. J. Hydrol., 513, 142-153, https://doi.org/10.1016/j.jhydrol.2014.03.052.,2014.

Paschalis, A., Fatichi, S., Molnar, P., Rimkus, S., and Burlando, P.: On the effects of small scale space-time variability of rainfall on basin flood response, J. Hydrol., 514, 313–327, https://doi.org/10.1016/j.jhydrol.2014.04.014, 2014.

---

## Author Comment (AC3) · 11 Jan 2021

**Answer to Peter Molnar (Short Comment 1)**

Thank you for your appreciation of our study and for sharing your ideas on our work and on hydro-sedimentary modeling in general. You are raising interesting questions and we agree that there is a high potential for future studies to apply physically based models with multiple sources to assess how structural and functional connectivity impact sediment dynamics at the outlet and to track sediment provenance.

Below, we give our thoughts to the points you raise. As you mentioned, the questions are interesting to the hydro-sedimentary modeling community in general. We don't claim that our answers are universally valid but reply to your questions with regard to our study and the context of our study sites.

1. Structural or functional connectivity by hydrology-sediment modelling?

First of all, we agree that functional connectivity and especially spatial and temporal rainfall variability is at least equally important as structural connectivity as a driver of sediment flux variability. Indeed, we think that the question how spatial variability of rainfall forcing impacts model output is a major knowledge gap in hydro-sedimentary modeling. While it is an active research topic in hydrological modeling (e.g. Emmanuel et al., 2015; Lobligeois et al., 2014), the question has not yet been properly addressed in hydro-sedimentary modeling. In that sense, the work of Battista et al. (2020a,b) is very relevant.

We also worked on functional connectivity ourselves, but to date this work is only published in Magdalena Uber PhD thesis (Chapter 5 of Uber 2020). A major finding was that temporal rainfall variability is a driver of within event variability of sediment source contributions and spatial rainfall variability strongly influences between event sediment dynamics.

Nonetheless, we are convinced that the location of the sources is very important and that there also is much to gain from studying structural connectivity separately. In your comment you write that "real basins never experience the kind of hypothetical climatic driving conditions studied by Uber et al., and runoff production in reality is heavily dependent on soil moisture that varies strongly in space and time". Here we slightly disagree. Of course the spatially uniform, synthetic triangular hyetograph that we apply is a simplification of reality, but it is not completely unrealistic. In both catchments some floods can be associated to widespread, stratiform precipitation (Hachani et al., 2017 for the OHMCV Observatory where the Claduègne catchment is located; Navratil et al., 2012 for the Galabre catchment) that is relatively uniform in space at the scale of our studied catchments (20 and 43 km$^2$). During these events occurring mainly in autumn and winter for which the exports of sediment are usually important, the rainfall amounts can be very high and spread over more than one day. As the soil moistures are high at these periods (e.g. Braud et al., 2014), the entire catchments are highly connected because overland flow occurs in widespread areas throughout the catchment. During such events the structural connectivity and particularly the location of the sources is very important as travel distance govern the sediment dynamics at the outlet.

However, we agree that structural and functional connectivity interact, particularly during more localized convective rainfall events, and one of our conclusions was that structural connectivity alone

can only explain a part of the observed variability of the source contributions. In the approach we chose, we gradually increased model complexity by firstly focusing on model sensitivity and structural connectivity and then including rainfall variability while keeping other parameters constant (work in progress and beyond the scope of the submitted manuscript).

You further write that you are "not convinced that event scale analyses and explorations of structural connectivity are helping us understand the processes better, unless we understand (and are able to model) why every event has a different hydrological, i.e. overland flow and therefore erosion, response across a catchment". Here, we focused on the event scale because at our study site the large majority of sediment export occurs during a small fraction of the time during extreme events. For example, for the Galabre catchment, Navratil et al. (2011) found that 90 % of the sediment was exported in 2 % of the time. This is why we focused on the event scale, but it is undoubtedly true that we are facing the problem of different initial conditions for different events. For future research we plan to include storage and remobilization of sediments in our model and to simulate chains of events.

2. How do we validate the hydrology-sediment models we use?

This is a very interesting question, thank you for sharing your ideas on the topic. We agree that overland flow is a crucial factor in rainfall-runoff-sediment transport modeling, but would like to extend the discussion to erosion and sediment transport. As many studies have shown, we are still not able to reliably reproduce observed sediment fluxes with physically based or empirical models (Jetten et al., 1999; Alewell et al., 2019). Furthermore, studies that combine sediment fingerprinting with erosion modeling have shown that even when modeled suspended sediment output was similar to measured one, this is not necessarily for the right reasons. E.g. Theuring et al. (2013) found that their model underestimated riverbank erosion and overestimated surface erosion, thus, the model predicts the right output but for the wrong reasons.

As demanded by other authors, this clearly shows that we need alternative strategies for model validation than the traditional comparison of modeled hydrographs and sedigraphs with observed ones at the outlet alone. We agree that there is much to gain from combining modeling and sediment fingerprinting and your work (Battista et al., 2020b) is a good example. This was also a motivation for our work. Furthermore, we also think that it is justified to bid farewell to the idea of exactly reproducing absolute fluxes at the outlet as proposed by Alewell et al. (2019):

"Nearing (2004) concluded that model validation is not just a matter of comparing measured to modelled data, one must also ask the question: 'How variable is nature?' We would like to add, that in bidding farewell to the idea of accurately predicting absolute values with models but rather concentrating on the prediction of relative differences, trends over times and systems reactions to processes and management practices, we can use models as tools to learn about the modelled systems and their reactions. In this conceptual approach, modelling in general and large-scale modelling specifically will per se not aim at an accurate prediction of point measurements."

In this sense, our aim was not to reproduce absolute liquid or solid discharge at the outlet but rather to understand the conditions (processes, forcing, model parameterization) that explain the observed patterns of sediment source contribution within and between events.

3. Identifying and tracking sediment sources?

We agree that it is very important to accurately map the locations of sediment sources and to include all potential sediment sources into the model and here we briefly give some thoughts on that topic that are based on our (previous) work. Again, we think that there is a high potential in combining modeling with sediment fingerprinting, especially at a high temporal resolution to gain insight in within

events and between events variability. E.g. the methodology proposed by Poulenard et al., 2012 and applied in Legout et al., 2013 and Uber et al., 2019 to quantify source contributions separately for each source can give us some confidence that we are not missing an important source. When the contributions of the individual sources in the mixing model are not forced to sum up to 100% but each sources contribution is determined independently, the simple calculation whether the contributions of all sources add up to approximately 100% is a simple test that can indicate problems in the fingerprinting protocol or hint at missing sources.

We would also like to stress that we did not use the model to identify the main sediment sources. This was achieved before with sediment fingerprinting (quantified source contributions were averaged over 11 floods in 7 years in the Claduègne catchment and over 77 flood events in 7 years in the Galabre catchment). This knowledge was imposed on the model by adjusting the erodibility coefficient for each source. In this way, we could use the model to go beyond the identification of main sources and use it for process understanding by tracking the sources and by analyzing the dynamics of their respective contributions to sediments at the outlet.

Regarding your concern that many events and longer time periods are needed to identify main sediment sources we fully share this view. Our work in the two catchments (not reported in the presented manuscript, but the fingerprinting studies by Legout et al., 2013 and Uber et al., 2019 as well as the modelling article including rainfall variability that we are currently working on) shows that sediment sources can differ considerably between events. This is in agreement with your conclusion in Battista et al., 2020b that the catchment can shift between different regimes.

In conclusion, we agree that there is a lot of potential for future studies to combine hydrosedimentary modeling with field data that can be used for source quantification with sediment fingerprinting. To our knowledge, past studies have mainly done so to identify main sediment sources within catchments (e.g. Palazon et al., 2016; Theuring et al., 2013; Wilkinson et al., 2013). This is certainly an excellent approach to validate model results and highly valuable for management purposes such as the question where to apply erosion control measures. However, several fingerprinting studies have shown important variability of source contributions within and between events (e.g. Evrard et al., 2011; Brosinsky et al., 2014; Vercruysse and Grabowski, 2019). Thus, physically based distributed models are excellent tools to understand this between and within event variability and we think that the studies of Battista et al.(2020b) as well as our study are interesting steps in this direction.

**References**

Alewell, C., Borelli, P., Meusburger, K., and Panagos, P. (2019). Using the USLE: Chances, challenges and limitations of soil erosion modelling. International soil and water conservation research.

Battista, G., Molnar, P., and Burlando, P. (2020a). Modelling impacts of spatially variable erosion drivers on suspended sediment dynamics, Earth Surf. Dynam., 8, 619–635, https://doi.org/10.5194/esurf-8-619-2020.

Battista, G., Schlunegger, F., Burlando, P., and Molnar, P. (2020b). Modelling localized sources of sediment in mountain catchments for provenance studies. Earth Surf. Process. Landforms, https://doi.org/10.1002/esp.4979.

Braud, I., Ayral, P.-A., Bouvier, C., Branger, F., Delrieu, G., Le Coz, J., Nord, G., Vandervaere, J.-P., Anquetin, S., Adamovic, M., Andrieu, J., Batiot, C., Boudevillain, B., Brunet, P., Carreau, J., Confoland, A., Didon-Lescot, J.-F., Domergue, J.-M., Douvinet, J., Dramais, G., Freydier, R., Gerard, S., Huza, J., Leblois, E., Le Bourgeois, O., Le Boursicaud, R., Marchand, P., Martin, P., Nottale, L., Patris, N., Renard,

B., Seidel, J.-L., Taupin, J.-D., Vannier, O., Vincendon, B., and Wijbrans, A. (2014). Multi-scale hydrometeorological observation and modelling for flash flood understanding. Hydrology and Earth System Sciences, 18(9):3733-3761.

Brosinsky, A., Foerster, S., Segl, K., López-Tarazón, J. A., Piqué, G., and Bronstert, A. (2014b). Spectral fingerprinting: characterizing suspended sediment sources by the use of VNIR-SWIR spectral information. Journal of Soils and Sediments, 14(12):1965-1981.

Emmanuel, I., Andrieu, H., Leblois, E., Janey, N., and Payrastre, O. (2015). Influence of rainfall spatial variability on rainfall-runoff modelling: Benefit of a simulation approach? Journal of Hydrology, 531:337-348.

Evrard, O., Navratil, O., Ayrault, S., Ahmadi, M., Némery, J., Legout, C., Lefèvre, I., Poirel, A., Bonté, P., and Esteves, M. (2011). Combining suspended sediment monitoring and fingerprinting to determine the spatial origin of fine sediment in a mountainous river catchment. Earth Surface Processes and Landforms, 36(8):1072-1089.

Hachani, S., Boudevillain, B., Delrieu, G., and Bargaoui, Z. (2017). Drop size distribution climatology in Cévennes-Vivarais region, France. Atmosphere, 8(12):233.

Jetten, V., De Roo, A., and Favis-Mortlock, D. (1999). Evaluation of field-scale and catchment-scale soil erosion models. Catena, 37(3-4):521-541.

Legout, C., Poulenard, J., Nemery, J., Navratil, O., Grangeon, T., Evrard, O., and Esteves, M. (2013). Quantifying suspended sediment sources during runoff events in headwater catchments using spectrocolorimetry. Journal of Soils and Sediments, 13(8):1478-1492.

Lobligeois, F., Andréassian, V., Perrin, C., Tabary, P., and Loumagne, C. (2014). When does higher spatial resolution rainfall information improve stream flow simulation? An evaluation using 3620 flood events. Hydrology and Earth System Sciences, 18(2):575-594.

Navratil, O., Esteves, M., Legout, C., Gratiot, N., Nemery, J., Willmore, S., and Grangeon, T. (2011). Global uncertainty analysis of suspended sediment monitoring using turbidimeter in a small mountainous river catchment. Journal of Hydrology, 398(3-4):246-259.

Navratil, O., Evrard, O., Esteves, M., Legout, C., Ayrault, S., Némery, J., Mate-Marin, A., Ahmadi, M., Lefèvre, I., Poirel, A., and Bonte, P. (2012). Temporal variability of suspended sediment sources in an alpine catchment combining river/rainfall monitoring and sediment fingerprinting. Earth Surface Processes and Landforms, 37(8):828-846.

Nearing, M. A. (2004). Soil erosion and conservation. In Wainwright, J. and Mulligan, M., editors, Environmental Modelling: Finding Simplicity in Complexity, chapter 22, p. 277-290. John Wiley & Sons, Ltd.

Palazón, L., Latorre, B., Gaspar, L., Blake, W. H., Smith, H. G., and Navas, A. (2016). Combining catchment modelling and sediment fingerprinting to assess sediment dynamics in a Spanish Pyrenean river system. Science of The Total Environment, 569-570:1136-1148.

Poulenard, J., Legout, C., Némery, J., Bramorski, J., Navratil, O., Douchin, A., Fanget, B., Perrette, Y., Evrard, O., and Esteves, M. (2012). Tracing sediment sources during floods using Di use Reflectance Infrared Fourier Transform Spectrometry (DRIFTS): A case study in a highly erosive mountainous catchment (Southern French Alps). Journal of Hydrology, 414-415:452-462.

Theuring, P., Rode, M., Behrens, S., Kirchner, G., and Jha, A. (2013). Identification of fluvial sediment sources in the Kharaa River catchment, northern Mongolia. Hydrological Processes, 27(6):845-856.

Uber, M. (2020). Suspended sediment production and transfer in mesoscale catchments : a new approach combining flux monitoring, fingerprinting and distributed numerical modeling. Dissertation, Université Grenoble Alpes, Grenoble, France. https://tel.archives-ouvertes.fr/tel-02926078

Uber, M., Legout, C., Nord, G., Crouzet, C., Demory, F., and Poulenard, J. (2019). Comparing alternative tracing measurements and mixing models to fingerprint suspended sediment sources in a mesoscale Mediterranean catchment. Journal of Soils and Sediments, pages 1-19.

Vercruysse, K. and Grabowski, R. C. (2019). Temporal variation in suspended sediment transport: linking sediment sources and hydro-meteorological drivers. Earth Surface Processes and Landforms.

Wilkinson, S. N., Hancock, G. J., Bartley, R., Hawdon, A. A., and Keen, R. J. (2013). Using sediment tracing to assess processes and spatial patterns of erosion in grazed rangelands, Burdekin River basin, Australia. Agriculture, Ecosystems & Environment, 180:90-102.

---

## Author Response (AR1)

Author's response to referee #1 comments on "How do modeling choices impact the representation of structural connectivity and the dynamics of suspended sediment fluxes in distributed soil erosion models?" by Uber et al.

In the following, the reviewer comments appear in black italic and our answers are provided in blue. When there are quotations from the text of the article, they appear in quotation marks.

We wish to thank the anonymous referee #1 for this very detailed and constructive review of our study and acknowledge the time spent and effort made. His/her comments helped us to substantially improve the paper and we hope that the changes made accordingly will contribute to an easier understanding of the text.

As a general response, we would like to point out that, upon reading many of the reviewer's comments or questions, we realized that the second objective of the article was mis-explained and therefore misunderstood. While the first objective is achieved by performing a sensitivity analysis of the choices made during the construction of the models (modelling scenarios 1, 2 and 3 in Table 2), the second objective corresponds to an opening towards the understanding of the temporal dynamics of fine sediment fluxes as a function of the geomorphological characteristics of two different catchments, in particular due to the location of the sources and their structural connectivity. Thus, the scenarios 4 described in Table 2 allow a better visualization and interpretation of the contributions of the different subcategories of sedimentary sources to the outlets.

In the revised version of the manuscript, we added, as proposed by referee #1, a study design section that links the different modeling scenarios in Table 2 to the two objectives reformulated in the table to be more explicitly linked to those announced in the introduction, also slightly reformulated.

After reading the comments of referee #1 we also became aware that the title of the manuscript only referred to our first objective while it did not refer to the second one. Thus we changed the title to the manuscript to "How do modeling choices and erosion zone locations impact the representation of connectivity and the dynamics of suspended sediments in a multi-source soil erosion model?"

*1 Summary*

*Uber et al. present a numerical modeling study that explores how modeling choices related to computational mesh generation, parameterization, and source-classification grouping influences a variety of output metrics describing hydrograph and sedigraph characteristics.* […]

Thank you for your general acknowledgement and positive feedback on our work.

*Below I describe comments and recommendations first in narrative form and then as line-level comments. My most substantial concern is that the paper lacks an overarching introduction to the study design—a section in which the authors set up the specific questions or hypothesis that they seek to address and connect them with a conceptual description of their numerical experiment design. A related comment is that I found the explanation of the modeling choices difficult to follow. Both of these issues meant that it was difficult to connect the study design and methods with the results and discussion.*

We addressed your concerns by including a short section "study design" as you proposed and made changes in the description of the modeling scenarios to better understand the modeling choices (see the further points).

*I recommend acceptance after major revisions and look forward to seeing this paper published.*

Thanks again for the constructive proposals and the recommendation for publication.

*2 Narrative Comments*
*2.1 Addition of an "Study Design" Section*
*The experimental design employed by the authors is valid and appropriate for the questions that they seek to pose. However, I found description clearly connecting the big picture questions ("what controls sediment flux from mesoscale watersheds") to the scenario design currently introduced by Section 3.4 and Table 2 was missing, or spread across too many sections of the paper.*
*I recommend that a new section be placed immediately after the introduction. In this section you would describe your experimental design and connect it to the big picture you have laid out in your introduction. Such a section would include the specific questions and hypotheses each scenario's experiment seeks to answer and an explanation of why this question was targeted.*
*While the reader may not know the details of the two sites or the model, your introduction should provide enough information such that this section can come before the more detailed methods section. Such a section will introduce to the reader the concrete questions your scenarios were designed to address.*
*Such as section should a description of the type of model analysis method used (e.g., a series of one-at-a-time sensitivity studies) and explain why this sort of method is appropriate to address the study objectives. Pianosi et al. (2016) is a good place to start for background on this topic. This will help the reader understand the type of results you will obtain.*
*In such a section, I would also like to see an introduction to why two catchments are used and why calculating whole-catchment connectivity metrics (described in Section 3.1); e.g., doing the same set of simulations across two catchments with different geology/land use/etc allows you to isolate how transferable your results are to catchments with different properties. This would also allow you to set up why you calculate a variety of catchment connectivity metrics (presented in Table 1) and explicitly state that you will eventually work to connect those connectivity metrics with the variability identified by the sensitivity analysis (a start at this is done at L461).*

We introduced a section "study design" as you proposed. However, we introduced it as an introduction of the modeling scenarios section, as it is directly linked to the description of the scenarios. The new section is now entitled "3.4 Study design and modeling scenarios". While this section is short we hope that the changes made in further sections will also help to improve the understanding of the study design. Thank you also for the recommendation of introducing our approach with the paper by Pianosi et al. and the hints to be more precise on the type of sensitivity analysis conducted.

*2.2 Improve explanation of modeling choices*

*The core of the study hinges on connecting the modeling set up described in Section 3.3 to the scenarios described in Section 3.4. However, I found it difficult to connect these two sections, mostly because I found it hard to follow exactly what the authors varied in their modeling set up.*

*The most constructive form of feedback I think I can provide here is a summary of what I understood after reading the paper four times, as well as what I would recommend so that I might have understood this after the first reading.*

Thank you for your summary from an outside perspective which helped us to be more precise on some parts, see comments below.

*Based on my reading, what I understand is that there Iber requires a computational mesh, and the mesh size can vary in space. Each mesh cell has a value for Manning's n and a value for α.*

*This is correct. We try to be more precise by changing the first sentence in section 3.3 that now reads "As a distributed model, Iber requires a computational mesh which is made up by three main modeling units with different spatial discretizations and roughness coefficients, i.e. the river network, the hillslopes and the badlands."*

*Choice 1: The considered area is divided up into three conceptual domains which influence the grid cell size and Manning's n value based on the CDA (hillslope, channel, badlands). Based on the delineation of these domains the mesh is discretized.*
*Next the mesh is parameterized with a spatially variable for Manning's n value. You might have chosen to let Manning's n vary smoothly, or something else, but you have chosen that the domain will get two Manning's values (channel and hillslope).*

This is correct. Again, we try to be more precise by adding "Values for Manning's and erodibility were assigned to each mesh element." in line 219 of the initially submitted version of the manuscript. We also added the following sentence in line 221: "It was chosen that the domain would get two Manning's values (channel vs hillslope), i.e a value for the modeling unit "river network" and another value for the modeling units "hillslopes" and "badlands".

*Choice2 focuses on those values. While water can fall on and run across the entire computational mesh, sediment can only be sourced from the bare bedrock areas. In these areas, the propensity to produce sediment is parameterized by α.*

We reformulated the sentence starting in line 222 which now reads "While runoff is generated and routed in the entire catchment, the production of sediment was limited to the potential erosion zones. The latter include all the mesh elements in the modeling unit "badland" and the mesh elements of the "hillslopes" modeling unit that belonged to the diffuse agricultural sources in the Claduègne catchment. The erosion zones were classified according to ..."

*I don't think the following was ever stated, but in order to produce the source proportion sedigraphs, I believe that some method of source tracking can be chosen in order to elucidate the dynamics of the basin.*

To be more precise about that, we reformulated the sentence starting in line 225 which now reads "Sediment production ($D_{rdd,s}$) was calculated in each mesh element of the potential erosion zones for each source class separately. Sediment transfer (Eq. 2) was then routed over the entire catchment.

Thus, separate sedigraphs for each source class were obtained at the outlet of the catchment and the contribution of each source class to total sediment flux could be calculated for every time step."

Furthermore, we thoroughly revised the description of the model in section 3.2 to be precise about this aspect. In Eq. 2 we added the subscript s to be more explicit about the fact that it was solved for each sediment class separately.

*Different classification of these tracked sources is represented by Choice 3 (I think). Thus Scenarios 2a–2d focus on Choice 1, Scenarios 3a–f focus on Choice 2, and I think that different delineations of source tracking (Choice 3), along with different choices for Manning's n yield Scenario 4.*

As mentioned in the general answer, the last set of scenarios (Sc. 4) were designed to answer the second objective written at the end of the introduction. The aim of Sc. 4 is to better interpret the modelled temporal dynamics of sediment fluxes for various groups of sediments depending on their geology and also on their distance to the outlet or to the river network. Thus, Sc. 4a and 4b do not really correspond to choices during modeling set up as the overall sedigraphs are not modified. They just allow a better visualization of the sediment origin (in subgroups) in order to facilitate the comparison with the connectivity indicators. To go further in the discussion and the interpretation of the impact of the location of sources within the catchment, and particularly to assess to which extent the conclusions derived from Sc. 4a and 4b were dependent on changes in roughness parameters, Sc. 4c and 4d were added, but they were initially not designed to be part of the sensitivity analysis conducted for objective 1.

*I would recommend the following to the authors:*
*Revise section 3.3 to describe more clearly what the modeling choices are such that they set the reader up to understand the details of scenario design discussed in the following section.*
*• In Section 3.3 or in the new "study design" section proposed above, explain why these choices are important to focus on. Are they the only choices? Are they the only ones which carry uncertainty? There are many things you might have focused on (e.g., assess the sensitivity to the channel grid cell size), but you chose these elements, why? To be clear, I think the elements you've chosen are great, I just want more description of why they were chosen.*

This suggestion was accepted and included in the new section "3.4 Study design and modeling scenarios" where it reads "Based on preliminary studies that are not reported here, these factors were found to be the most important ones in determining sediment flux dynamics. While other factors (erodibility, rainfall intensity) crucially influence absolute values of erosion and suspended sediment concentration, their values are less important to determine arrival times and temporal dynamics of source contributions."

*• Clarify how the source classification is represented in model specification. Does this choice not influence the model physics, but just the model output that permits a different view on the dynamics?*

Yes, this is the case. E.g. the sedigraphs of the sources "Limestone 1" (close to the outlet) and "Limestone 2" (further) in scenario 4a sum up to the sedigraph of the source "Limestone" (which includes close and distant subsources) in the basic scenario. Thank you for pointing out that this was not clear. The following sentence was added at the end of section 3.4 "It should be stressed that this source classification does not influence model physics, i.e. total sediment yield from a source (close + distant sources) remains the same as in the basic scenario where they are not differentiated."

*• Explain why sediment is only sourced from the bare bedrock.*

We changed the sentence starting in line 126 to "The land use is dominated by forests and scrublands, which are permanently covered by vegetation and are thus assumed to be negligible as sediment sources. Agricultural zones are barely present in the catchment."

*2.3 Improve connection between study design and discussion*

*The structure of the discussion roughly follows the three non-base case scenarios and presents the most salient aspects of the results. However, within each of the major discussion sections, I found the text difficult to follow. I suspect that by being more explicit about the target questions and hypotheses earlier in the text the authors will be able to very lightly restructure the discussion such that the reader is easily able to connect the discussion with the study intent and numerical experiments.*
*In addition, the end of the discussion starts to tie together the basin-scale metrics presented in Table 1 and the numerical modeling results. It would be beneficial to introduce earlier on that you will do this and describe in more detail how this is accomplished (e.g., regression, rank correlation). Knowing that this sort of analysis is coming will help explain why all of the basin-scale metric are calculated and discussed starting*
*at L136.*

Thank you for that remark. We included this idea in the new section "3.4 Study design and modeling scenarios" by adding "[…] indicators of structural connectivity of the two catchments are used to describe the configuration of sediment sources in the catchment. They are compared to the modeled hydro-sedimentary fluxes both qualitatively by visual analyses and quantitatively by means of the calculation of characteristic times of the hydrographs and sedigraphs (e.g. time of concentration, lag time)"
We prefer not to use a specific term like regression or rank correlation because we are comparing only 5 data points at a maximum (4 sources in the Galabre catchment + liquid discharge)

*2.4 Figures*
*The interactive figures provided by Uber et al. (2020) are a fantastic complement to the paper. I might consider adding catchment as a facet (e.g., facet grid with scenario catchment) because this would facilitate comparison between catchments.*

This is a nice idea, but it is not easy to implement. We prefer to keep the interactive figures as they are.

*I'd also like to applaud your consistency in the use of color to denote geological unit across figures. This should be a standard expectation, but it isn't, and it makes comprehension much better.*

Thank you very much for the positive feedback on the (interactive) figures.

*My primary concern with figures relates to the maps presented in Figure 1. This figure shows us inconsistent information across the two catchments (e.g., badlands only shown in 1a)*

We revised figure 1 in a way that consistent information is shown for the two catchments. The land use information in fig 1b was omitted as it is indeed essential for this study and it is presented by Esteves et al. (2019). Now the figure shows the erosion zones that were considered in the two catchments as colored patches.

[Figure]

*and does not show us all of the information used in the modeling study that is the focus of the work. I recommend that Figure 1 be redrafted into a series of rows that shows the reader the main elements used in model initialization for each catchment. For example, row one might show a shaded relief map with the river system and badlands areas, row 2 would show the considered geologic units used, row 3 might show the weighting factor W presented by Borselli et al. (2008), while row 4 would show the roughness based weighting factor of Cavalli et al. (2013).*

We prefer to keep figure 1 simple as the paper already has several figures that are composed of different subfigures. The information you request is contained in Magdalena Uber's PhD thesis available at https://tel.archives-ouvertes.fr/tel-02926078 and a reference was added at the end of the caption of figure 1: "Further maps of the study sites can be found in Uber (2020)".

*2.5 Code availability*

*For the purposes of computational reproducibility, state the version of Iber used.*

We changed the introduction of the model at the beginning of the section "3.2 Model description" to be clearer about the fact that we worked with a version of the model that is in development: "Surface runoff, soil erosion and sediment transport in the study catchments were modelled with an ad-hoc version of the software Iber (Bladé et al., 2014) developed in a previous study by the authors (Cea et al. 2016)". While the hydraulic model can be downloaded from the iberaula website, the erosion and sediment transport module is still a research version developed initially by Cea et al. (2016) which cannot be downloaded yet.

*No statement has been made about model input file availability. Such files should be digitally archived for the purpose of reproducibility.*

Given that the erosion and transport part of the model cannot be downloaded yet, we do not think there is any interest in dropping the input files on a repository.

*3 Line Level Comments*

*Bullet points in this Section indicate "<LineNumber>", "F<Figure Number>", or "T<Table Number>".*

*36 The term "Mediterranean and mountainous" is used a few times, first here. Mediterranean could be interpreted a few ways: e.g., places with a Mediterranean climate, places near the Mediterranean. Recommend being more specific about what is meant.*

Thank you for pointing that out. We meant the terms as "having a Mediterranean and mountainous climate". However, this is not the case in the study by Vanmaercke et al. that was cited in line 36, so we prefer to clarify it in line 42 where we replace "Mediterranean and mountainous watersheds" with "watersheds with a Mediterranean or mountainous climate".

*56 Recommend giving an example of your objectives and thus how structural connectivity is represented to anchor this abstract concept on a concrete example or two.*

We replaced the sentence ending in this line by the sentence "In the context of soil erosion and sediment transfer studies it is of interest how active erosion zones are linked to the catchments outlet." to be more precise about the use of the concept of structural connectivity in this study.

*76 I suspect the sentence that ends in this line needs a reference.*

We added the reference "(Merrit et al., 2003)"

*87 Be more specific about which models and provide examples with associated references.*

The half-sentence "such as WEPP (Laflen et al., 1991)), Kineros (Woolhiser et al., 1990)) and Mike 11 (Hanley et al., 1998)" was added.

*100 Additional subsubsection headers would have helped me understand this section more easily. For example Section 3.1 discusses both a description of the catchments and connectivity metrics calculated, and Section 3.3 discusses many different aspects of the model set up. I would split each of these subsections into multiple subsubsections.*

We agree with the reviewer for section 3.1. Thus we split it in a first subsection labelled "Catchment descriptions" and a second one labelled "Connectivity indicators". However, we decided to keep section 3.3 unchanged.

*129 A few lines or a paragraph summarizing the similarities and differences of the two catchments would benefit the reader here.*

We agree to sum up the main differences at the end of the paragraph, by adding the sentences "In comparison, the Galabre catchment is smaller and steeper than the Claduègne catchment. The distribution of the erosion zones differs in the two catchments, with the ones in the Galabre catchment being more dispersed over the entire catchment but smaller in size due to the absence of diffuse agricultural sources." Their main similarity is the fact that they are both mesoscale catchments in a mountainous and Mediterranean context. As this is stated several times before we prefer not to repeat it again here.

*136 Some statements about why these connectivity metrics were chosen would benefit the reader.*

This was explained in lines 140-142 "The distance to the outlet and the distance to the stream of a given position in the catchment serve as proxies of longitudinal (upstream-downstream) and lateral (hillslope-channel) connectivity in the sense of Fryirs (2013)" and in lines 144 – 146 "However, neither of these measures takes into account surface roughness and slope. Thus, two of the most widely used indicators of connectivity, i.e. the IC proposed by Borselli et al. (2008) and the adjusted version of IC proposed by Cavalli et al. (2013), were calculated." We hope that adding the precisions in the brackets helps to better understand the explanations to the reader who might not be familiar with the work by Fryirs (2013).

*In addition, explain (here or in something like the proposed "Study Design" section) what you expect to learn from these metrics and how they are used.*

Ok, we included that explanation in the section "3.4 Study design and modeling scenarios" as you proposed.

*137 The distance to the outlet metric has been called the "width function" by the landscape evolution modeling community Hancock et al. (2010, 2002). Work by this community has shown that it is not a particularly good metric for comparing catchment topography, but is a does provide a good assessment of hydrology. It may be useful to connect with this literature.*

Thank you for the hint and the recommendation of the reference. We added the sentence "The distance to the outlet metric refers to the width function applied as a measure of network structure and catchment shape by Hancock et al. (2010).".

*138 Mathematically represent the connectivity indices of Borselli et al. (2008) and Cavalli et al. (2013) here so that the reader can more clearly understand what they represent.*

We prefer to refer the reader to the original publication here and not go into too much detail. The two indices are not the most important metrics used in this paper. It is already a bit unusual to describe the calculations of these metrics in the study sites section but we took that decision in order to keep it short. Thus, going into further detail would be beyond the scope of this short description of the metrics.

*171–173 This detail of model set up should be located elsewhere. Probably is a subsubsection of Subsection 3.3 (see also the comment at L237 and 289.*

We agree and relocated the sentence in line 237.

*211 Being able to connect this discussion of badlands in model set up to a consistent picture of where badlands are located is why I mentioned earlier that Figure 1 should be revised to include consistent information about each catchment.*

Ok, see our response to your comment on figure 1.

*215 Connect and justify the choice of a 5 m minimum grid size with relevant field observations and the numerics of the Iber model? E.g., how does this compare with the range of values for channel width in each catchment? Do the numerics of Iber benefit from a relationship between minimum grid cell size and channel width (e.g., smallest grid cell = channel width, 10 grid cells = channel width).*

Given that the same surface water and sediment routing equations are applied in all three units (the river network, the hillslopes and the badlands), the model presents a continuous representation of hillslopes and the river network. In order that the river flow strictly follows the slope of the topography, we had to choose a cell size that is in the order of magnitude of the resolution of the DEM (1 m). A smaller mesh size of 1 m for example would strongly increase the number of mesh elements and thus computation time without increasing the accuracy of the results so this value is a compromise between exact representation of the topography, computational efficiency and accuracy of results. Thus, the minimum grid size of 5 m was chosen as a compromise between the representation of the flow structure in the river, computation time and accuracy of results.

*217 20 m seems like a rather large grid cell size for gullied areas. Explain and/or justify this value.*

You are right that the topography on the steep badlands is not exactly reproduced by this value. Again, it presents a compromise between detail, computational efficiency and accuracy of results. We did preliminary analyses that are not reported in this paper on the impact of the mesh size (only for hydrology and on a subcatchment) by conducting a convergence-of-the-mesh experiment: starting at a coarse mesh size and then gradually decreasing it. At some point the results converged, i.e. a smaller mesh size did not lead to significantly different results. This is how the optimal mesh size was determined. The resulting optimal mesh size of 20 m for badlands is related to the fact that erosion is represented only by the detachment of rainfall which is modelled in a simple way as a function of the rainfall amount. This optimal mesh size would have been different if detachment by overland flow had been implemented as the topography controls the water heights and velocities.

*222 The erosion source locations should be shown in Figure 1 in addition to the subplots shown in later figures.*

We revised figure 1 so that it now shows the erosion zones clearly.

*222 If I'm interpreting this correctly, I believe you are saying that sediment production can only occur in the areas of bare bedrock. This should be explained further and justified. In addition, discuss how this model set up decision impacts the implications of this study for overall soil erosion (as these bare bedrock patches only make up a small portion of the study watershed).*

Thank you for the remark. We remind that the erosion zones were previously defined in the sediment fingerprinting studies by Legout et al. (2013) and Uber et al. (2019) for the Galabre and Claduègne catchment respectively. In line 126 we noted that in the Galabre catchment the land use classes other than the badlands are "permanently covered by vegetation and are thus assumed to be negligible as sediment sources". Badlands are therefore the only sources of erosion. In the Claduègne catchment, apart from badlands, some diffuse, agricultural sources have to be considered. In the new version of the manuscript, we have stressed such a difference between the two catchments following your comment on line 129 by adding the sentence "The distribution of the erosion zones differs in the two catchments, with the ones in the Galabre catchment being more dispersed over the entire catchment but smaller in size due to the absence of diffuse agricultural sources". Also, we hope that with changes made to figure 1 it is now easier to see the extent and location of the erosion zones.

*227–236 It is difficult to understand if this section of text is summarizing the work of Uber et al. (2019) or if it is presenting an analysis of modeling results. Revise to clarify this point.*

We rephrased the two sentences ending in line 235: "$SSY_{s,ev}$ is the contribution of source s to $SSY_{ev}$ and was calculated based on the mean source contributions. They were estimated with sediment fingerprinting in the Claduègne catchment by Uber et al., 2019 and in the Galabre catchment by Legout et al., 2013." We hope that in this way it gets clear that the reference Uber et al., 2019 refers only to the sediment fingerprinting in the Claduègne catchment. The rest of the section explains the calculations made for this study.

*227 Introduce the units of α when the variable is first presented.*

Thank you for pointing it out. We state the unit in line 230 where the formula is given now.

*237 No discussion of time discretization, model run duration, or external forcing (e.g., rain) is present in the prior subsection. These elements of model set and running should be discussed.*

Based on your comment above we move the description of the hyetograph (rainfall forcing) here. Further we added "The simulated time is 24 h, including 12 h of rain and 12 h for the fluxes to reach the outlet" (line 295) to be precise about the model run duration here. The description of the model (section 3.2) was revised thoroughly and now states the method of time discretization: "The solver is explicit in time, meaning that the maximum time step that can be used to evolve the equations in time is limited by the Courant-Friedrichs-Lewy (CFL) condition (Courant et al. 1967). This implies that the time step in typical applications is of the order of one second or less. The CFL condition is implemented in the solver and thus, the computational time step is automatically evaluated from the grid size, water velocity and water depth"

*237 Based on the results presented, it appears that Iber has the capability of tracking the source of water/sediment as it moves through the catchment and that how these source regions are grouped is what is meant by the "source classification" column of Table 2. This aspect of the model should be discussed. As best as I can tell this is a critical aspect of interpreting Scenario 4.*
*In addition, it is not clear whether this choice of model set up impacts the dynamics of water and sediment (or if it just impacts how they are analyzed). E.g., are simulation 1 and 4a and 4b the same simulation just analyzed/post processed differently?*

We hope that this gets evident after our general answer at the beginning of this document and the changes we made following your earlier comments in narrative form.

In line 278 it now says: "It should be stressed that this source classification does not influence model physics, i.e. total sediment yield from a source (close + distant sources) remains the same as in the basic scenario where they are not differentiated." Further in line 225, it now says "Sediment production ($D_{rdd,s}$) was calculated in each mesh element of the potential erosion zones for each source class separately. Sediment transfer (Eq. 2) was then routed over the entire catchment. Thus, separate sedigraphs for each source class were obtained at the outlet of the catchment and the contribution of each source class to total sediment flux could be calculated for every time step." Eq. 2 was also changed to be more explicit that it was solved for each class separately.

*260 The simulations of Scenario 3 represent two one-at-a-time sensitivity studies (Sc. 3a–3c for sensitivity to hillslope Manning's n and Sc. 3d–3f for channel). Recommend using more formal language to describe the numerical experiments as it will help the reader anticipate the type of results presented.*

We stated that in the new section "3.4 Study design and modeling scenarios" you proposed earlier and repeat it in line 256 by adding one-factor-at-a-time sensitivity analysis in brackets: "We tested the impact of varying the CDA threshold on the modeled hydro-sedimentary response while keeping all other parameters unchanged compared to the basic scenario (one-factor-at-a-time sensitivity analysis)".

*268 It is not clear to me how the different options for source classification of Scenario 4 relate to changes in the parameterization of the model. Were different values of α used? Something else? Clarify.*

There are no changes in the parameterization of the model. We hope our general answer and specific response to your comment on line 278 allow a better understanding about the aims of Sc. 4. For example, in the Claduègne catchment, the difference is that instead of having three source classes in the basic scenario (badland, basaltic, sedimentary), there are 6 source classes (badlands-close, badlands-distant, basaltic-close, basaltic-distant, sedimentary-close, sedimentary-distant) in scenario 4b and 4d. This is visualized in figures 10 and 11. We hope the changes made as explained in our response to your narrative comments and the one on line 278 make this source classification and its implication easier to understand.

*In addition, these scenarios include two options for the Manning's n value, the base case and one in which the hillslope value is low and the channel value is high. The results of Scenarios 4c and d are discussed at L454. Formally introduce what the purpose of this sub-scenario is.*

Thank you for pointing that out. We added at the end of section 3.4 the sentences "Besides the values for Manning's n used in the basic scenario, in Sc. 4c and 4d we used values for Manning's n that were less contrasted between the hillslopes and the river network. This was done to assess whether the interpretation of Sc. 4a and 4b (i.e. the discussion on how the location of the sources in terms of their distance to stream or outlet, impacts the temporal dynamics of SS fluxes at the outlet) depended on the values of n.".

*272–274 This sentence, in which you link the changes to the model set up with a hypothesis is exactly the sort of text that a "Study Design" section would benefit from. Recommend that similar sentences for each scenario exist and be present in such a section.*

Following this comment, we made sure that for every Scenario a sentence like this explain why this scenario was created. For scenario 2 we added line 258: "Thus, it can be assumed that modeled sediment dynamics are sensitive to this parameter.". For scenario 3 we think the explanation is already in the text l.261: "As the first objective of this study is to assess the impact of choices made during model set-up on the simulated sediment flux dynamics, the model was run with different values of Manning's n in the river network modeling unit on one hand and in the hillslopes and badlands modeling units on the other hand".
In the section "3.4 Study design and modeling scenarios", it is now stated: "The underlying hypothesis is that both modeling choices (notably CDA threshold and Manning's n) and catchment characteristics (structural connectivity of the sources) determine travel times from the sources to the outlet. With the presented study design, it could be assessed whether modeling choices or actual catchment configurations were more important in generating output variability".

*280 This section clearly describes what model output metrics were used, however it does not explain why these output metrics were chosen or justify why they are appropriate given the overall goals of the study. This section should be expanded to include this information.*

Thanks for pointing that out. We added the following sentence at the end of Section 3.5: "We use these metrics to quantitatively assess differences in model output between the scenarios described above."

*289 This sentence describing model run details should go elsewhere in the text. Probably in a section on external forcing, along with the text currently located at L171–173 (see comment at L237).*

We have completed the information on the duration of the simulations at the end of section 3.3 but we have left this sentence in this section as we have not found a better place.

*296 Be more specific about which aspects of the model. Some aspects are sensitive and some are not.*

We changed "the model was sensitive" to "modeled hydrographs and sedigraphs were sensitive".

*307 Connect this statement with new text earlier in the paper describing why two catchments are used. Set the reader up for this sort of discussion by explaining why two catchments are used, and comparing/contrasting them.*

We hope that the introduction of the modeling scenarios with the new section "3.4 Study design and modeling scenarios" allows to better understand the interest of studying two catchments. Particularly the following sentence was added to: "With the presented study design, it could be assessed whether modeling choices or actual catchment configurations were more important in generating output variability."

*313 Justify why this is a reasonable interpretation and connect with literature.*

We do not have found any relevant study to cite for this purpose. However, analyzing all the characteristics of both catchments leads to a clear contrast of their slopes. Whatever the compartments (hillslopes, intermittent streams and main stream) the slopes are on average two to three times higher in the Galabre than in the Claduègne catchments, leading to modelled hydrological response times smaller in the Galabre than in the Claduègne catchment in accordance with measurements.

*337 This statement presents a different conclusion than Table 3 and the text near L296 which states that different CDA values result in output metric variability. These three elements of results and discussion should be consistent.*

The emphasis here is on "in this range". We rephrased it so that it becomes more evident: "Overall, our results showed that the thresholds of 15, 35 and 50 ha produced very similar results. Thus, in this range, the model was not very sensitive to the CDA threshold."

*344-350 The purpose and reasoning of the argument you advance here is not clear. As you highlight it in the conclusion (L487) I believe you think it is an important point. Recommend this text be revised.*

Thank you for pointing out that the paragraph was not clear, we rephrased it: "This result showed that it is important to use a CDA threshold that is in the same order of magnitude as the value that produces a realistic river network. Field observations or detailed maps (i.e. topographic map at scale 1:25000) can be valuable sources of information for this purpose. The sensitivity of model output to variations of the CDA threshold was also observed by other authors (Pradhanang and Briggs, 2014). For our modeling set-up it is reassuring that model results converged when the CDA threshold used is derived from field observations."

*352 The section of Table 2 that shows the results of Scenario 3 indicates that changing Manning's n in the hillslope has a larger impact on the results than changing the channel value. This should be discussed.*

It is true that generally changing n on the hillslopes has a larger impact than changing n in the river network. But this might not be true universally. Thus, we prefer to keep the formulation as it is ("Interestingly, in the Claduègne catchment liquid discharge was more sensitive to changes in $n_{hillsl.}$ than to $n_{river}$ while solid discharge was more sensitive to $n_{river}$. This was not the case in the Galabre where both liquid and solid discharges were more sensitive to $n_{hillsl.}$", line 360). Actually, changing n on the hillslopes had less impact on the sedigraphs than what could be expected. We discuss that in the paragraph l.378-381 where we have added information in brackets in the new version of the manuscript: "Our results showed that even though modeled liquid discharges were sensitive to $n_{hillsl.}$(e.g. maximum liquid discharge changed by 24% in the Claduègne catchment and 12% in the Galabre catchment), the sedigraphs of the main sources and thus of total suspended solid discharge were much less sensitive to this parameter (maximum solid discharge changed by 3% in the Claduègne catchment and by 1% in the Galabre catchment, Figure 8). This was due to the fact that in both catchments the main sediment sources were located close to the river (Table 1, Figure 2). Thus, only a small fraction of the trajectory of particles was located on the hillslopes."

*372 What is meant by "more stable"?*

We added "more stable in time" to be more precise.

*379 Here and elsewhere, sensitivity should be presented as a relative measure. E.g., this output was more sensitive to choice/parameter A than to choice/parameter B. Without the comparison the statement is uninterpretable.*

We added the percent change with respect to the basic scenario as a quantitative measure of sensitivity (information in brackets): "Our results showed that even though modeled liquid discharges were sensitive to $n_{hillsl.}$,(e.g. maximum liquid discharge changed by 24% in the Claduègne catchment and 12% in the Galabre catchment), the sedigraphs of the main sources and thus of total suspended solid discharge were much less sensitive to this parameter (maximum solid discharge changed by 3% in the Claduègne catchment and by 1% in the Galabre catchment , Figure 8)"

*392 Here you discuss both a contrast between the two catchments, the analysis of Scenario 4, and connecting basin-wide metrics of IC with the sensitivity results. Recommend structuring the section to help the reader anticipate this.*

We hope that the clarification made on objective 2 help the reader to better anticipate what is compared and discussed in this section.

*393 Introduce this idea in the study design.*

As you recommended, we announced the comparison of the two catchments in the new section "3.4 Study design and modeling scenarios": "With the presented study design, it could be assessed whether modeling choices or actual catchment configurations were more important in generating output variability."

*397–399 This has already been stated.*

Thank you for pointing that out. We propose to delete the sentence "The rising limb of the hydrograph was also steeper in the Galabre than in the Claduègne catchment  (shorter Tlag and Tc, Figure 5, Table 3)." However we prefer to keep the second sentence. The steeper slopes of the Galabre catchment are assumed to be the reason for several findings: the faster reaction of the catchment, the steeper hydrograph and sedigraph, the lower sensitivity to Manning's n in the river.

*402 Add a figure reference.*

The figure reference is given 3 lines above: "From Figures 7 and 9 a general pattern of the contribution of the different geological sources to total solid discharge can be derived: In the Claduègne catchment […]" To make it more evident that this paragraph refers to figures 7 and 9 we propose to replace the full stop with a colon in line 400.

*407 More specific. E.g., close = first, or something different?*

We added a complement to the sentence (the last part of the sentence after the last comma): "In the Galabre catchment at the onset of the event ("1"), suspended sediment originated almost entirely from the black marls, i.e. the source closest to the outlet."

*421 It is not clear if Scenario 4 represents a different approach to tracking something else? Because the description of how Sc. 4 was constructed is incomplete it is nearly impossible to understand the results of Sc. 4.*

Thank you for pointing out that the description of Sc. 4 was insufficient to understand it from an external perspective. We hope that this gets clearer after the changes we made in the methods section according to your comments above. However, as it seems to be an important point, we added a further explanation on how results were obtained l.422: "In this way, model output consisted of separate sedigraphs for the close and distant subsources of a given source class. The sum of these sedigraphs is the same as the sedigraph of that source class in the basic scenario."

*423–425 Give the reader a little more context about "typical interpretations of discharge sediment flux hysteresis" and provide a description of what a clockwise vs counterclockwise loop means.*

We expanded the paragraph by giving a short description of the interpretations of Q-SSC flux hysteresis: "Figures 10 and 11 showed for the Galabre catchment that the limestone sources that were close to the river and the ones that were close to the outlet exhibited a clockwise discharge-sediment flux hysteresis pattern while the distant ones exhibited an anticlockwise pattern. These results confirmed typical interpretations of hysteresis loops, i.e. the assumption that clockwise loops indicate a dominance of close sources because maximum sediment flux occurs before peak discharge while anticlockwise hysteresis patterns indicate a dominance of more distant sources (Bača, 2008; Misset et al., 2019). The results further highlighted that the sedigraphs of the different sediment sources were strongly related to their location in the catchments and their structural connectivity."

*431 Not sure what is meant by this sentence.*

We are not sur which sentence is referred to. We rephrased the two sentences which now say "Thus, the mean distance to the outlet was not sufficient to determine travel times of the sources to the outlet. Additionally, the triangular rain applied to both catchments lasted had a rather long periodduration, much longer than the times of concentration of both catchments."

*448 Unclear if distance to the outlet (or stream) being considered is related to the parameterization or the analysis of the results.*

The latter is the case. The sentence was rephrased accordingly: "When the results were analyzed in terms of the distance to the outlet, it was remarkable that […]"

*461 This sentence starts a new line of inquiry: which basin-wide metrics (Table 1) best predict the sensitivities documented by the numerical experiments. A more explicit discussion of the methods used here (e.g., comparing basin wide metrics to sensitivity ranking) should be added to the methods. In addition, the description of this analysis should be expanded.*

We understood that this comment is related to the comment above that the reviewer wished to have a more explicit statement of the method used to "correlate" basin metrics to the metrics of the sedigraph. However, as stated earlier, we wish to refrain using statistical terms such as correlation or rank analysis for the comparison of only 5 data points.

*465 This sentence is not clear.*

Thank you for pointing that out. The idea behind this sentence is explained in the following sentences so we deleted this unclear sentence.

*468 It is not clear that your study design supports this type of analysis. To my ability to tell you have not varied the location and/or erodibility of the sediment sources within the catchment. As such, your study design does not permit assessment of how variability in location of sediment sources influences the output metrics.*

Indeed we cannot prove this statement with quantitative metrics of sensitivity. Nonetheless, we think that the analysis is justified. We did not vary locations of the sources but we compared different sources with different locations. Concerning erodibility, it is true that we don't report on how changes made in the erodibility coefficient impacts model output. This is due to the fact that detachment rate is linearly related to erodibility in our model. Thus, changing the values of alpha changes absolute values of detachment rate but not the temporal dynamics of sediment fluxes. We stressed that following your earlier comments by adding "While other factors that were not considered here (erodibility, rainfall intensity) crucially influence absolute values of erosion and suspended sediment concentration, their values are less important to determine arrival times and temporal dynamics of source contributions" in the new section "3.4 Study design and modeling scenarios".

*469 The point you are making here is not clear, mostly because the text introduced at L344-350 is not clear.*

Thank you again for noticing that this point was not clear. We hope that the changes we made in the results section (former L344-350) make it easier to follow this conclusion.

*478 Unclear how the study is about source soils when the only erodible material is the exposed bedrock. This should be addressed here and earlier in the text.*

We changed "source soils" to "sources" here. We also revised the description of what was considered a source in section 3.3: "[…] the potential erosion zones. The latter include all the mesh elements in the modeling unit "badland" and the mesh elements of the "hillslopes" modeling unit that belonged to the diffuse agricultural sources in the Claduègne catchment". Furthermore, Figure 1 now shows clearly what was considered as a source in the two catchments (Badlands in the Galabre catchment, Badlands as well as cultivated soils in the Claduègne catchment).

*Most Figures In the many multi-panel plots I recommend use of consistent x and y axis limits and/or explicit notation of inconsistent axis limits in Figure captions.*

Whenever this was possible we used consistent x and y limits. However, whenever two erosion zones were compared, it was not possible because then the dynamics in the graphs of the less erosive zone would not be visible because of the very different erodibility of the sources (e.g. the y-axis of fig. 6). Furthermore, we focus on temporal dynamics and not on absolute values in this study. Thus, we did not state this explicitly in the figure legends.

*F10–13 The panel (f) is the sort of information that would be great to have in a revised Figure 1. The background color scheme for the inset maps (distance to outlet, distance to stream) should be represented by a legend.*

As noted above, we prefer to keep figure 1 simple to stress the most important information on the location of the erosion sources and wish to keep the panel (f) in these figures where the focus is on the distance to the outlet and distance to the stream metrics.

*T2 The layout of the table makes it difficult to see the difference between the scenario 4 options.*

We revised the column "Aim" in Table 2 to better relate this table to the 2 objectives of the study. In the text we better explained why two sets of values for n were used in Sc. 4 following your comment above.

*T3 1. Why are the simulations used for Scenario 4 not in the table?*

As the classification of the sources was different in Sc. 4 than in the other scenarios we would have to give all 3 metrics ($T_{lag}$, $T_c$, $T_{spr}$) for each one of 31 subsources so this would add nearly 100 lines to the table which is already quite long.

*2. Recommend adding some vertical lines to help guide the viewer in separating*
*Sc. 1, Sc. 2, and the two halves of Sc. 3.*

We prefer to keep the classic table layout without vertical lines.

*3. Overlaying the table text on top of a tile plot is a great addition. However, the darkest blue values make reading the text impossible.*

We changed the text color to white so that it is easier to read the text on the darkest blue shades.

*4. Not clear why some values have NA, explain.*

Following your comment, we explained this in the caption of the table: "NA values indicate that the hydrograph or sedigraph did not recede to 0.1 Qmax within the simulated time."

Author's response to Editor G. Hancock comments on "How do modeling choices impact the representation of structural connectivity and the dynamics of suspended sediment fluxes in distributed soil erosion models?" by Uber et al.

In the following, the reviewer comments appear in black italic and our answers are provided in blue. When there are quotations from the text of the article, they appear in quotation marks.

We wish to thank you for your comments that helped us to substantially improve the paper and we hope that the changes made accordingly will contribute to an easier understanding of the text.

*Review of 'How do modelling choices impact the representation of structural connectivity and the dynamics of suspended sediment fluxes in distributed soil erosion models' by Uber et al. This is a timely paper. Given the number of hydrology and sediment transport models available understanding the sensitivity of parameters is extremely important. Therefore, the topic is of high interest. The paper reports on an assessment of model sensitivity in two catchment in France. The field data and numerical experiment is nicely done. However, there a few comments that need to be addressed that can make the paper stronger.*

Thank you very much for the review of our manuscript and for the recognition of our work.

*The Abstract summarises the paper nicely. However, the Introduction needs some attention. At the end of the Introduction, I largely agree and understand all the you have described, but I am not sure where the paper is really going. I have read the Introduction several times and it is not clear what you are really going to do. This leads to a comment about Section 3.4 (and its logic) which is somewhat difficult to rationalise in terms of the various model runs and setup. The Introduction needs to be refocussed with a much stronger and defined aim particularly at the end of the section. The sentence on lines 72-74 seems to summarise the overall intent of the paper. While the sentences on lines 92-94 are quite vague.*

Thank you for pointing out that the introduction was not clear and that the objectives were not easily understandable. This flaw also got evident from some of the comments of the anonymous referee #1 and to some misunderstandings of the referee despite considerable effort made and multiple readings of the paper.

Following your comment and the comments by referee #1 we reformulated the sentence in line 92-94 you refer to: "This paper contributes to improve our understanding of the hydrosedimentary processes in the catchment that lead to sediment flux variability at the outlet". We also slightly reformulated the objectives "Since model outputs are supposed to be highly sensitive to the choices made during model set-up, the first objective is to assess the impact of the choices made during model discretization and parameterization on modeled suspended sediment flux dynamics. A second objective is to assess how structural connectivity, particularly the location of the sediment sources, impacts modeled suspended sediment flux dynamics for both catchments."

Moreover, we propose to change the title to better reflect these two objectives: "How do modeling choices and erosion zone locations impact the representation of connectivity and the dynamics of suspended sediments in a multi-source soil erosion model?"

We further revised the column "Aim" in table 2 to better relate this table to the two objectives of the study.

*Line 174- Soil erosion module I have no problem with using a single layer in an instance like this. However, the model used here only models erosion? No deposition? I realise that the inclusion of deposition adds complexity and would likely slow model run time but what is the effect of neglecting this on the findings? Landscape Evolution Models have demonstrated that including deposition has a significant influence on erosion particularly gullying. I say this as you mention gullies in the Badlands in Section 3.3.*

It is true that we don't include deposition in our model and we agree that it could be considered as a strong simplification of reality. However, in both catchments, the slopes of the stream are high (>2.5%) and mainly incised into the bedrock. Contrary to what can happen downstream of the measuring stations where the slopes of the river decrease considerably, the temporary storage of fine sediments and their resuspension are not dominant processes compared to the fluxes of fine sediments coming from the primary sources of the catchments. For further studies we plan to include deposition and resuspension to assess to which extent these temporary storages are important processes to consider in such catchment configuration. Nonetheless, in this first step, we wished to keep the model as simple as possible and to focus on the processes that we believed were the most important ones in our catchments (i.e. rainfall detachment and transport via surface runoff). Both of our study sites are prone to heavy rainfalls and flash floods that lead to high sediment exports during these events. We focus on these events where we believe that the sources are highly connected to the river network.

*A further issue is that you are only modelling suspended sediment? Is this the case? What about bedload? Is the quantity of bedload significant? Should you be examining total load? Line 420-424. Here you talk about total solids. Does this include bedload? Or is it suspended load?*

You are right, we are only modeling suspended sediments. When we wrote "total solid discharge" we meant the sum of solid discharge from the different sources. It is true that this is ambiguous, so we changed it to "total suspended load" or to "total suspended solid discharge" in line 421 and elsewhere.

*Conclusion. Can this be rewritten to summarise succinctly the interesting work here. A Conclusion should summarise and largely be standalone with data presented. I suggest that lines 489-492 have been discussed elsewhere. As presented it reads like an extension of the Discussion and does not do the paper justice.*

As suggested we have reorganized and shortened the conclusion to highlight the main findings of this study. We therefore propose the following conclusion in the revised version of the article that will be submitted.

"This study aimed to improve our understanding of hydrosedimentary processes leading to temporal variability in the contribution of potential sources to suspended sediments at the outlet of two mesoscale catchments using a distributed, physically based numerical model. As a first objective, we analyzed to which extent the choices made during model discretization and parameterization impacted the modeled suspended sediment flux dynamics. The shape and the magnitude of the modeled hydrographs and sedigraphs were sensitive to the contributing drainage area threshold to define the river network and to Manning's roughness parameter n in the river network and on hillslopes. However, the model was less sensitive to all three values once the parameters varied only in a restricted, reasonable range. The pattern of modeled source contributions remained relatively similar when the CDA threshold was restricted to the range of 15 to 50 ha, n on the hillslopes to the range 0.4-0.8 and to 0.025-0.075 in the river.

Then, the second objective was to assess how the location of geological sources in the catchment impacted the modelled temporal dynamics of suspended sediments at the outlets. The classification of the geological sources in subgroups showed that the hydrosedimentary responses differed in the two studied catchments due to the combined effects of the distance from the sources to the point of entry of sediments in the river network, the distance of the sources to the outlet as well as the slopes of hillslopes and rivers. Among the various structural connectivity indicators tested to describe the geological sources, the mean distance to the stream was found to be the most relevant proxy of the temporal characteristics of the modeled sedigraphs."

*Other issues:*

*Line 128. What is 'molasses'?*

It is a geological classification of sedimentary rocks. This was given in line 123 "The catchment is entirely located on sedimentary rocks comprising limestones (34%), marls and marly limestones (30%), gypsum (9%), molasses (9%) and Quaternary deposits (18%)."

*I really liked the interactive figures*

Thank you for the positive feedback on the interactive figures.

[revised manuscript text omitted]

It allowed to assess how structural connectivity in the catchments governs hydrosedimentary fluxes at the outlet. On the one hand,to which extent the modeling choices made during model discretization and parameterization could impacted the representation of the structural connectivity in the model two mesoscale catchments and thus determines travel times and modeled hydrographs and sedigraphs. On the other hand, structural connectivity is governed by the location of the sources in the catchment, the distance from the sources to their point of entry in the river network, their distance to the outlet as well as slopes on the hillslopes.
[revised manuscript text omitted]
| Distance to stream $[km]$ | $0.44 \pm 0.35$ | $0.21 \pm 0.19$ | $0.67 \pm 0.34$ | $0.42 \pm 0.36$ | $0.53 \pm 0.37$ | $0.89 \pm 0.47$ | $0.39 \pm 0.35$ | $0.34 \pm 0.24$ | $0.57 \pm 0.35$ |
| $IC$ (Borselli et al., 2008) | $-9.18 \pm 0.61$ | $-8.35 \pm 0.43$ | $-9.30 \pm 0.37$ | $-8.75 \pm 0.66$ | $-8.84 \pm 0.75$ | $-7.94 \pm 0.39$ | $-7.95 \pm 0.60$ | $-8.19 \pm 0.36$ | $-8.03 \pm -0.42$ |
| $IC$ (Cavalli et al., 2013) | $-5.85 \pm 0.53$ | $-5.50 \pm 0.34$ | $-6.34 \pm 0.50$ | $-5.73 \pm 0.50$ | $-4.56 \pm 0.50$ | $-4.52 \pm 0.33$ | $-4.57 \pm 0.55$ | $-4.81 \pm 0.35$ | $-4.56 \pm 0.40$ |
| **Erodibility** | | | | | | | | | |
| Suspended sediment yield $[t\,y^{-1}]$ | 15947 | 12394 | 1084 | 2469 | 12856 | 953 | 1956 | 7474 | 2473 |
| Specific yield $[t\,km^{-2}y^{-1}]$ | 380 | 38623 | 2087 | 589 | 666 | 2780 | 2113 | 57075 | 7418 |
| Rain erodibility $\alpha^{b)}$ $[g\,mm^{-1}m^{-2}]$ | 3.1 | 37.5 | 2.0 | 0.6 | 7.4 | 2.8 | 2.1 | 57.1 | 7.4 |

**Table 1:** Characteristics of the two catchments and the erosion zones. KG is Gravelius' compactness indicator defined as the ratio between the catchment perimeter (P) and the one of a circle with equal surface. The values given for the slopes on the hillslopes, the distance to the outlet, the distance to the streams and the two connectivity indicators (IC) represent the mean +/- standard deviation. The mean slopes in the river network are given for the entire network including intermittent streams (defined with a threshold of CDA of 15 ha) and for the main, perennial network (CDA of 500 ha). a) The values correspond to the slope in the river network on the basaltic plateau and on sedimentary geology and are not limited to the erosion zones. b) Rainfall erodibility corresponds to the mass of sediment detached on 1m² by 1mm of rain (Cea et al., 2015).

| Sc. | Th_CDA [ha] | Source classification | $n_{river}$ [-] | $n_{hillsl.}$ [-] | Aim |
|---|---|---|---|---|---|
| 1 | 15 | Geology | 0.050 | 0.8 | Basic scenario |
| 2a | 35 | Geology | 0.050 | 0.8 | |
| 2b | 50 | Geology | 0.050 | 0.8 | Impact of modeling choice for the river network threshold(spatial discretization) on the temporal dynamics of SS fluxes |
| 2c | 150 | Geology | 0.050 | 0.8 | |
| 2d | 500 | Geology | 0.050 | 0.8 | |
| 3a | 15 | Geology | 0.050 | 0.2 | |
| 3b | 15 | Geology | 0.050 | 0.4 | |
| 3c | 15 | Geology | 0.050 | 0.6 | Impact of modeling choice for the parameterization of (Manning's nroughness) on the temporal dynamics of SS fluxes |
| 3d | 15 | Geology | 0.025 | 0.8 | |
| 3e | 15 | Geology | 0.075 | 0.8 | |
| 3f | 15 | Geology | 0.100 | 0.8 | |
| 4a | 15 | Geology and distance to the outlet | 0.050 | 0.8 | |
| 4b | 15 | Geology and distance to the stream | 0.050 | 0.8 | Impact of the location of erosion zones within the catchments on the temporal dynamics of SS fluxesDynamics between more and less connected sources |
| 4c | 15 | Geology and distance to the outlet | 0.100 | 0.2 | |
| 4d | 15 | Geology and distance to the stream | 0.100 | 0.2 | |

**Table 2:** Model scenarios (Sc) detailed according to the value of the contributing drainage area threshold to define
the river network (ThCDA), the approach to classify the sources, the values for Manning's roughness parameter
(n) in the river network and on the hillslopes and the aim of the respective scenario.

| | 1 Basic Scenario | 2a Th$_{CDA}$ = 35 ha | 2b Th$_{CDA}$ = 50 ha | 2c Th$_{CDA}$ = 150 ha | 2d Th$_{CDA}$ = 500 ha | 3a $n_{hillsl.}$ = 0.2 | 3b $n_{hillsl.}$ = 0.4 | 3c $n_{hillsl.}$ = 0.6 | 3d $n_{river}$ = 0.025 | 3e $n_{river}$ = 0.075 | 3f $n_{river}$ = 0.100 |
|---|---|---|---|---|---|---|---|---|---|---|---|
| **Claduègne** | | | | | | | | | | | |
| $T_{lag,Q_l}$ [h] | 4.00 | 4.33 | 4.50 | 5.33 | NA | 2.67 | 3.17 | 3.67 | 3.50 | 4.50 | 5.00 |
| $T_{c,Q_l}$ [h] | 5.67 | 6.33 | 6.67 | 9.33 | NA | 3.17 | 4.00 | 4.83 | 4.67 | 6.50 | 7.33 |
| $T_{spr,Q_l}$ [h] | 12.33 | 12.67 | 13.00 | 15.33 | NA | 10.67 | 11.17 | 11.67 | 11.83 | 12.67 | 13.17 |
| $Q_{l,max}$ [m$^3$s$^{-1}$] | 41.65 | 40.16 | 39.14 | 32.91 | 22.14 | 51.44 | 48.00 | 44.57 | 42.51 | 40.67 | 39.64 |
| $Q_{s,max}$ [kg s$^{-1}$] | 191.04 | 198.67 | 183.24 | 169.41 | 108.65 | 197.45 | 201.52 | 196.98 | 163.88 | 217.06 | 230.97 |
| $T_{lag,Q_s}$ bad [h] | 2.67 | 2.83 | 3.00 | 3.67 | 6.00 | 1.83 | 2.17 | 2.50 | 2.17 | 3.17 | 3.67 |
| $T_{c,Q_s}$ bad [h] | 3.00 | 3.00 | 3.33 | 4.50 | 9.33 | 2.33 | 2.50 | 2.83 | 2.67 | 3.33 | 3.67 |
| $T_{spr,Q_s}$ bad [h] | 9.17 | 9.00 | 9.17 | 10.00 | 14.67 | 9.50 | 9.17 | 9.17 | 9.67 | 8.83 | 8.50 |
| $T_{lag,Q_s}$ bas [h] | 6.17 | 6.67 | NA | NA | NA | 3.67 | 4.83 | 5.50 | 5.50 | NA | NA |
| $T_{c,Q_s}$ bas [h] | 10.83 | 11.17 | NA | NA | NA | 5.50 | 7.50 | 9.17 | 9.00 | NA | NA |
| $T_{spr,Q_s}$ bas [h] | 16.00 | 15.83 | NA | NA | NA | 12.17 | 13.50 | 14.67 | 14.83 | NA | NA |
| $T_{lag,Q_s}$ sed [h] | 3.83 | 4.17 | 4.33 | 4.83 | NA | 2.17 | 2.83 | 3.50 | 3.50 | 4.17 | 4.33 |
| $T_{c,Q_s}$ sed [h] | 7.17 | 7.83 | 8.17 | 8.83 | NA | 3.00 | 4.67 | 6.00 | 6.67 | 7.50 | 7.67 |
| $T_{spr,Q_s}$ sed [h] | 14.00 | 14.50 | 14.83 | 15.33 | NA | 10.67 | 12.00 | 13.00 | 14.17 | 13.83 | 13.67 |
| **Galabre** | | | | | | | | | | | |
| $T_{lag,Q_l}$ [h] | 2.33 | 2.67 | 2.83 | 3.67 | 4.67 | 1.33 | 1.67 | 2.00 | 2.17 | 2.50 | 2.67 |
| $T_{c,Q_l}$ [h] | 2.67 | 3.33 | 3.67 | 5.33 | 7.50 | 1.83 | 1.83 | 2.17 | 2.33 | 3.00 | 3.17 |
| $T_{spr,Q_l}$ [h] | 10.83 | 11.33 | 11.50 | 12.83 | 14.50 | 10.33 | 10.50 | 10.50 | 10.83 | 10.83 | 10.83 |
| $Q_{l,max}$ [m$^3$s$^{-1}$] | 22.71 | 21.83 | 21.50 | 19.47 | 17.89 | 25.38 | 24.43 | 23.58 | 22.79 | 22.61 | 22.54 |
| $Q_{s,max}$ [kg s$^{-1}$] | 95.70 | 94.73 | 94.29 | 103.65 | 69.15 | 96.64 | 95.15 | 94.54 | 94.08 | 97.66 | 99.52 |
| $T_{lag,Q_s}$ li [h] | 3.67 | 4.33 | 4.50 | 5.50 | NA | 2.00 | 2.67 | 3.33 | 3.50 | 4.00 | 4.17 |
| $T_{c,Q_s}$ li [h] | 6.00 | 7.83 | 8.17 | 10.83 | NA | 2.50 | 3.67 | 4.83 | 5.50 | 6.50 | 7.00 |
| $T_{spr,Q_s}$ li [h] | 14.00 | 16.17 | 16.00 | 17.17 | NA | 11.33 | 12.00 | 13.00 | 13.67 | 14.17 | 14.33 |
| $T_{lag,Q_s}$ ma [h] | 1.83 | 2.17 | 2.17 | 2.67 | 5.33 | 1.17 | 1.33 | 1.67 | 1.67 | 2.00 | 2.17 |
| $T_{c,Q_s}$ ma [h] | 2.67 | 3.00 | 3.33 | 4.17 | 10.17 | 1.67 | 2.00 | 2.33 | 2.33 | 3.00 | 3.17 |
| $T_{spr,Q_s}$ ma [h] | 11.17 | 11.33 | 11.67 | 12.33 | 18.17 | 11.17 | 11.00 | 11.00 | 11.33 | 11.33 | 11.50 |
| $T_{lag,Q_s}$ mo [h] | 1.83 | 1.83 | 2.00 | 2.67 | 3.83 | 1.17 | 1.33 | 1.50 | 1.50 | 2.00 | 2.17 |
| $T_{c,Q_s}$ mo [h] | 2.33 | 2.50 | 2.50 | 3.00 | 7.50 | 1.67 | 1.83 | 2.17 | 2.00 | 2.67 | 2.83 |
| $T_{spr,Q_s}$ mo [h] | 10.33 | 10.33 | 10.17 | 10.17 | 13.33 | 10.33 | 10.17 | 10.33 | 10.50 | 10.00 | 10.00 |
| $T_{lag,Q_s}$ qu [h] | 2.67 | 3.17 | 3.33 | 3.50 | 5.83 | 1.50 | 2.00 | 2.33 | 2.50 | 2.83 | 3.17 |
| $T_{c,Q_s}$ qu [h] | 4.00 | 5.00 | 5.00 | 5.67 | 8.67 | 2.17 | 2.83 | 3.50 | 3.67 | 4.33 | 4.67 |
| $T_{spr,Q_s}$ qu [h] | 12.00 | 12.67 | 12.67 | 12.67 | 14.83 | 10.83 | 11.17 | 11.67 | 11.83 | 11.83 | 11.83 |

| Change [%] | 0-9 | 10 - 19 | 20 - 29 | 30 - 49 | 50 - 69 | 70 - 89 | 90 - 119 | 120 - 149 | 150 - 179 | ≥ 180 |
|---|---|---|---|---|---|---|---|---|---|---|

**Table 3:** Calculated characteristics of modeled hydrographs and sedigraphs for the different scenarios. Abbreviations: $T_{lag;Ql}$: lag time of liquid discharge, $T_{c;Ql}$: time of concentration of liquid discharge, $T_{spr;Ql}$: spread of the hydrograph, $Q_{l;max}$: peak of liquid discharge. $Qs$ refers to solid discharge and the characteristic times are calculated for each source separately (i.e. badlands, basaltic and sedimentary in the Claduègne catchment; limestone, black marl, molasses and quaternary deposits in the Galabre catchment). The background color of the cells represents the percent change of each value with respect to the basic scenario. NA values indicate that the hydrograph or sedigraph did not recede to 0.1 $Q_{max}$ within the simulated time.